



# 1  A partially-coupled hydro-mechanical analysis of the Bengal Aquifer
# 2  System under hydrological loading

Nicholas D. Woodman[1†], William G. Burgess[1], Kazi Matin Ahmed[2] and Anwar Zahid[3]
[1]Department of Earth Sciences, University College London, London WC1E 6BT, UK, [2]Department of Geology, Dhaka
University, Dhaka 1000, Bangladesh, [3]Bangladesh Water Development Board, Dhaka, Bangladesh, [†]current address: Faculty
of Engineering and the Environment, University of Southampton, Southampton SO17 1BJ, UK
*Correspondence to*: Nicholas D Woodman (n.d.woodman@soton.ac.uk)
**Abstract.** The coupled poro-mechanical behaviour of geologic-fluid systems is fundamental to numerous processes in
structural geology, seismology and geotechnics but is frequently overlooked in hydrogeology. Substantial poro-mechanical
influences on groundwater head have recently been highlighted in the Bengal Aquifer System, however, driven by terrestrial
water loading across the Ganges-Brahmaputra-Meghna floodplains. Groundwater management in this strategically important
fluvio-deltaic aquifer, the largest in south Asia, requires a coupled hydro-mechanical approach which acknowledges poro-
elasticity. We present a simple partially-coupled, one-dimensional poro-elastic model of the Bengal Aquifer System, and
explore the poro-mechanical responses of the aquifer to surface boundary conditions representing hydraulic head and
mechanical load under three modes of terrestrial water variation. The characteristic responses, shown as amplitude and phase
of hydraulic head in depth profile and of ground surface deflection, demonstrate (i) the limits to using water levels in
piezometers to indicate groundwater recharge, as conventionally applied in groundwater resources management; (ii) the
conditions under which piezometer water levels respond primarily to changes in the mass of terrestrial water storage, as applied
in geological weighing lysimetry; (iii) the relationship of ground surface vertical deflection to changes in groundwater storage;
and (iv) errors of attribution that could result from ignoring the poroelastic behaviour of the aquifer. These concepts are
illustrated through application of the partially-coupled model to interpret multi-level piezometer data at two sites in southern
Bangladesh. There is a need for further research into the coupled responses of the aquifer due to more complex forms of
surface loading, particularly from rivers.

## 26  1 Introduction

Throughout the Bengal Basin, the floodplains of the Ganges, Brahmaputra and Meghna (GBM) rivers (Fig. 1) are underlain
by the Bengal Aquifer System (BAS), the largest aquifer in south Asia and the source of water to over 100 million people
(Burgess et al., 2010). Management of the BAS groundwater resource relies on monitoring water levels in networks of





observation boreholes, taking the conventional approach that changes in groundwater heads represent volumetric changes in
groundwater storage through recharge and drainage (Shamsudduha et al., 2011). This approach presumes the hydraulic
behaviour of the aquifer to be decoupled from its mechanical response to changes in stress. Recently, however, the distinctively
poroelastic behaviour of the BAS has been recognised (Burgess et al., 2017), by which groundwater heads are subject to
substantial mechanical perturbation driven by changes in the mass of terrestrial water storage (TWS) above the surface of the
aquifer. A coupled hydro-mechanical approach is necessary for understanding groundwater conditions and managing resources
in this environment, particularly in relation to recharge (Shamsudduha et al., 2012), sustainability of groundwater abstraction
for irrigation (Shamsudduha et al., 2008) and municipal water supply (Ravenscroft et al., 2013), and the security of schemes
for mitigation against groundwater arsenic (Michael and Voss, 2008) and salinity (Rahman et al., 2011; Sultana et al., 2015).
The generally coupled poro-mechanical nature of geologic-fluid systems is well-established (Neuzil, 2003); porewater
pressures affect the stress state and vice-versa. These interactions are accepted as important where groundwater conditions are
related to faulting (Roeloffs, 1988; Rojstaczer and Agnew, 1989; Sutherland et al., 2017), earthquakes (Manga et al., 2012),
pumping-induced aquitard responses (Verruijt, 1969), ground subsidence (Burbey et al., 2006; Erban et al., 2014), glacial
loading effects (Bense and Person, 2008; Black and Barker, 2016) and surface water interactions (Acworth et al., 2015; Boutt,
2010). Use of ground surface vertical displacements to infer aquifer or groundwater conditions (Chaussard et al., 2014; Reeves
et al., 2014) is also predicated on coupling of the hydraulic and mechanical behaviour of aquifer sediments. For simulation of
transient groundwater flow in aquifers, however, a decoupling simplification is frequently applied such that the elastic equation
does not need to be solved simultaneously. Thus, the flow equation is solved without consideration of internal stresses and
strains or mechanical boundary conditions. Despite this, the poro-mechanical nature of confined aquifers is embedded in the
concept of specific storage which incorporates the elastic compressibility of the aquifer materials (Domenico and Schwartz,
1998; Green and Wang, 1990; Narasimhan, 2006). The decoupling assumption is reasonable where the effects of mechanical
loading can be considered insignificant, either when the changes in load are small, or when the applied load is mostly borne
by the solid rather than the fluid (Black and Barker, 2016). Neither of these conditions apply to the BAS sediments, which are
highly compressible (Steckler et al., 2010) and subject to substantial and extensive TWS mechanical loads due to heavy rainfall,
deep flooding and large river discharges as a consequence of the annual monsoon (Shamsudduha et al., 2012).
In the event of laterally-extensive changes to mechanical loads and/or hydraulic heads above the surface of an aquifer, and
laterally-homogeneous aquifer properties, by symmetry it may be deduced that lateral strains are zero. This condition gives
rise to a *partial* coupling of the elastic and fluid pressure equations (Neuzil, 2003). In the case of *partial* coupling, changes to
the mechanical load due to the changing mass of water near or at the surface may be included within the flow equation, one-
dimensionally in the vertical direction, and the solutions will satisfy all the equilibrium and compatibility requirements for
stress and strain. There is no need to solve the elastic equation in order to calculate pressures in the aquifer, although once the
flow equation is solved, the pressures can be substituted into the elastic equation to provide stresses and strains. A sub-set of
this partially-coupled condition occurs where there is negligible groundwater flow, due to very low hydraulic gradients, low
permeability or a combination of both. This can be the situation in extensive fluvio-deltaic aquifers of low topographic relief



such as the BAS (Burgess et al., 2017) if mechanical loading is imposed at the surface in a manner which does not induce
significant vertical hydraulic gradients. Under these conditions, porewater pressures are determined by changes to surface
mechanical loading alone, and changes in groundwater head may be taken as a measure of changes in TWS mechanical loading
above the surface of the aquifer. This is the conceptual basis for geological weighing lysimetry (Bardsley and Campbell, 2007,
1994; van der Kamp and Schmidt, 1997, 2017) as used in diverse environments to determine ΔTWS at the scale of individual
catchments (Barr et al., 2000; Lambert et al., 2013; Marin et al., 2010; Smith et al., 2017). Geological weighing lysimetry has
been suggested as suitable for mapping the variability of ΔTWS within the Bengal Basin (Bardsley and Campbell, 2000;
Burgess et al., 2017), complementary to basin-scale estimates based on the Gravity and Climate Recovery Experiment
(GRACE) satellite mission (Shamsudduha et al., 2012; Tapley et al., 2004; Tiwari et al., 2009).
The purpose of this paper is to explore the behaviour of the BAS as a poroelastic aquifer subject to a variety of extensive TWS
mechanical and hydraulic loads. For this, we treat separate components of TWS across the GBM floodplains as *inundation*
(free-standing surface water such as paddy, floods, beels, and ponds), *unconfined storage* (water in the unsaturated zone and
in saturated pores in the intermittently saturated zone of the aquifer), *elastic storage* (water in the saturated pores in the
permanently saturated zone), and *rivers* (surface water flowing in rivers and drainage channels). Processes that alter the TWS
loads include rainfall and evaporation, rising and falling river stage, flooding and drainage of the land surface, varying soil
moisture storage and a fluctuating water table. Groundwater pumping modifies the water balance and induces additional hydro-
mechanical responses. These processes differ in their timing, the geometry of the TWS stores they affect and the relationship
between their resultant hydraulic and mechanical expressions. First, we apply the concept of *partial* coupling to seek
characteristic responses of the aquifer to extensive TWS loads originating as (a) surface water inundation, (b) water table
fluctuation and (c) water bodies hydraulically isolated from the aquifer. These loading styles are examined with and without
pumping. The results address important questions for the BAS which are likely also relevant to similarly extensive and
strategically important fluvio-deltaic aquifer systems elsewhere in south Asia (Benner et al., 2008; Fendorf et al., 2010; Larsen
et al., 2008; Tam et al., 2014; Xu et al., 2011): how can piezometer heads in the poroelastic aquifer be used to indicate recharge,
as required for conventional groundwater resources management; under what conditions can piezometer heads be used to
measure ΔTWS using geological weighing lysimetry; can ground surface deflections be related to changes in groundwater
storage; and what errors may arise if the poroelastic behaviour of the aquifer is ignored? Second, we apply the partial coupling
approach to these questions in the BAS, with reference to multi-level piezometer data from Khulna and Laksmipur in southern
Bangladesh (Fig. 1).

## 92  2 Methods

We firstly set out the partially-coupled 1D poromechanical approach that we use to examine the implications of specific surface
(upper boundary) loading scenarios, with aquifer parameters set to represent the BAS underlying the GBM floodplains (Fig.
1). We consider an equivalent homogeneous uniform medium, as well as a layered structure based on lithological sections.



The results provide a diagnostic framework which we apply to analysis of loading styles at Khulna and Laksmipur in southern
Bangladesh.

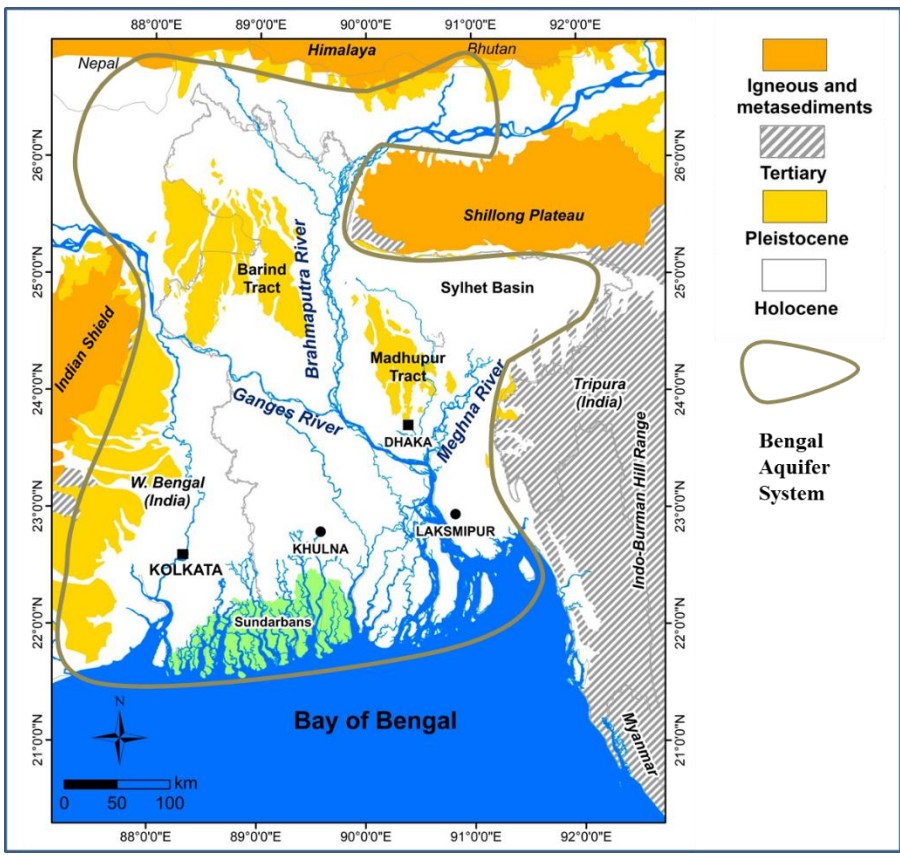

**Figure 1. Location map showing the extent of the Bengal Aquifer System (BAS) and the Ganges-Brahmaputra-Meghna (GBM)**
**floodplains.**



## 2.1 Poromechanical equations

We concentrate on the isothermal coupling between water flow and the elastic behaviour of the BAS sediment, and assume that the aquifer material behaves in a linear-elastic way. This is likely to be reasonable under repeated mechanical load-unload cycles, provided there is no secular decline in groundwater level sufficient to cause effective stress to exceed the previous loading maximum.

The governing equations for elastic deformation of a porous solid can be derived from the constitutive equations for stress, force equilibrium and strain compatibility. In 3D, the poro-elastic constitutive relations between elastic stress and strain are the same as the classical relationships for an elastic solid coupled to the pore-pressure by Terzaghi's effective stress law (Neuzil, 2003):

$$\sigma_{ij} = 2G\varepsilon_{ij}\delta_{ij} + 2G\frac{v}{1-2v}\varepsilon_{kk}\delta_{ij} + \alpha_B p\delta_{ij} \tag{1}$$

where, $\delta_{ij}$ is the Kronecker delta (which is zero when $i \neq j$ and one when $i = j$) and following the Einstein Summation convention; stresses ($\sigma_{ij}$) and strains ($\varepsilon_{ij}$) are positive in compression; $p$ is the porewater pressure (Pa), $v$ is Poisson's ratio (-), $G$ is the shear modulus (MPa), and $\alpha_B = 1 - K/K_s$, where, $K$ (MPa) is the bulk modulus of the porous medium and $K_s$ (MPa) is the bulk modulus of the solid grains. Here we assume that the solid grains are effectively incompressible ($K_s \gg K$) and hence $\alpha_B = 1$.

Equation (1) can be simplified to 1D where there is a uniform mechanical load with wide lateral extent such that there are no lateral strains. The medium is considered to sit on a rigid base, with the top surface free to move, so strain can only be vertical, thus:

$$\sigma_{zz} = K'\varepsilon_{zz} + \alpha_B p \tag{2}$$

where,

$$K' = 3K(1-v)/(1+v) \tag{3}$$

and the bulk modulus, $K$ and shear modulus, $G$ are related to Young's modulus $E$ by $K = \frac{E}{3(1-2v)}$ and $G = \frac{E}{2(1+v)}$. Just as the elastic equations have a pore pressure term, the isothermal, Darcian groundwater flow equation contains a coupled stress term (Neuzil, 2003):

$$\nabla \cdot \kappa(\nabla p + \rho g \nabla z) = S_{s3}\frac{\partial p}{\partial t} - S_{s3}\beta\frac{\partial \sigma_t}{\partial t} - gJ \tag{4}$$

where $\kappa$ is the hydraulic conductivity (m s⁻¹), $p$ is the pore pressure (Pa), $z$ is the elevation (m), $J$ is a source term used here to simulate groundwater abstraction by pumping and $\sigma_t = (\sigma_{xx} + \sigma_{yy} + \sigma_{zz})/3$ (Pa).

Changes to $\sigma_t$ (here termed 'mechanical loads') are applied as a boundary condition at the surface, and are transmitted by the solid skeleton to the entire solid at the acoustic velocity. This represents partial 'coupling'; if there are no internal loads applied and provided the changes to the surface load are known, then the flow equation can be solved without a need to solve the elastic equations. Deformations can be found from Eq. (2), in conjunction with the compatibility relationships.



The 3D specific storage is defined as:

$$S_{s3} = \rho g \left[ \left( \frac{1}{K} - \frac{1}{K_s} \right) + \left( \frac{n}{K_f} - \frac{n}{K_s} \right) \right] \tag{5}$$

where $n$ is the porosity, and $K_f$ is the modulus of the water (MPa). The (3D) loading efficiency, or Skempton's coefficient, $\beta$,
is defined as:

$$\beta = \frac{\left( \frac{1}{K} - \frac{1}{K_s} \right)}{\left( \frac{1}{K} - \frac{1}{K_s} \right) + \left( \frac{n}{K_f} - \frac{n}{K_s} \right)} \tag{6}$$

In the event of uniform areal mechanical loading, and where lateral strains are negligible, Eq. (4) simplifies to 1D:

$$\nabla \cdot \frac{k \rho g}{\mu} (\nabla p + \rho g \nabla z) = S_s \frac{\partial p}{\partial t} - S_s \xi \frac{\partial \sigma_{zz}}{\partial t} - gJ \tag{7}$$

where $\xi = \beta(1 + \nu)/[3(1 - \nu) - 2\alpha\beta(1 - 2\nu)]$ is the one-dimensional loading efficiency and $S_s$ is the one-dimensional
specific storage(van der Kamp and Gale, 1983)

$$S_s = S_{s3}(1 - \lambda\beta) \tag{8}$$

where $\lambda = 2\alpha_B(1 - 2\nu)/3(1 - \nu)$.
We therefore consider a simplified system: a 1D column of aquifer with no-flow boundaries on the sides and base, and no
horizontal strain (Fig. 2). On the upper boundary, the changing TWS is simulated by means of a changing head and a changing
mechanical load, according to the nature of the contributing hydrological components. Under this simplification, vertical
displacement at the surface will arise in only two ways: by contraction or expansion of the pore space where there is a net
change in the volume of water in the column, and by contraction or expansion of the pore water. Being limited to 1D movement,
these volume changes are entirely taken up by vertical displacement.
The reference frame is the base of the model which is assumed fixed in space and set at 1 km depth, acknowledging the
variation in aquifer thickness between south-east Bangladesh, 3000 m (Michael and Voss, 2009b) and West Bengal, 300 m
(Mukherjee et al., 2007). Within this domain, equations (2) & (7) are solved analytically for a homogeneous uniform material
in the absence of pumping, and numerically where layers of individually homogeneous materials are simulated, with and
without pumping. Where pumping is simulated, the water is assumed to be taken uniformly from the pumping-interval. For
simplicity, earth-tides are neglected.
**2.2 Analytical solution**
Taking Eq. (7) and assuming homogeneous $K$, $E$ and that $J = 0$ , converting $p$ to metres head, $h$ (i.e. $h = \rho g p + z$), and $\sigma_t$ to
metres of load (i.e. $L = \sigma_t/\rho g$, where $\rho$ (kg m$^{-3}$) is the density of water and $g$ (m s$^{-2}$) is the acceleration due to gravity)
(Burgess et al., 2017; van der Kamp and Schmidt, 1997) gives:

$$D \frac{\partial^2 h}{\partial z^2} = \frac{\partial h}{\partial t} - \xi \frac{\partial L}{\partial t} \tag{9}$$


where 1D hydraulic diffusivity is defined as $D = \frac{k\rho g}{\mu S_s}$.
Applying the following sinusoidal hydraulic and mechanical loading boundary conditions to Eq. (9) where we introduce
parameter, $\alpha$, which can be set to zero to give the case of a load in the absence of a varying head, and otherwise is kept at 1:

$$h(0,t) = H(t) = \alpha H_0 \cos(\omega t) \tag{10}$$

$$L(t) = S_y H_0 \cos(\omega t)$$

The following solution is obtained:

$$h(z,t) = \alpha B \cos(\omega t - \psi) \tag{11}$$

where $\psi$ is the lag (in radians) behind the head $H(t)$ and mechanical loads $L(t)$ at the boundary and:

$$B = \sqrt{\gamma^2 + 2\gamma(\alpha - \gamma)e^{-\theta}\cos(\theta) + (\alpha - \gamma)^2 e^{-2\theta}} \tag{12}$$

$$\psi = \tan^{-1}\left(\frac{(\alpha - \gamma)\sin(\theta)}{(\alpha - \gamma)\cos(\theta) + \gamma e^{\theta}}\right)$$

$$\theta = z\sqrt{\frac{\omega}{2D}} = z\sqrt{\frac{\pi}{DT}} \quad \text{and} \quad \gamma = S_y \xi$$

157         In the event that the mechanical load, L, is negligible compared to applied head H (e.g. where either $S_y$ is very small

or $\xi$ is very small), the hydraulic-only solution is well known (van der Kamp and Maathuis, 1991):

$$h(z,t) = H_0 \exp(-\psi)\cos(\omega t - \psi) \tag{13}$$

where the lag is now $\psi = \theta$. Thus, the lag increases with depth or with increasing forcing frequency and the amplitude
decreases exponentially with $\theta$.
Displacement and change in groundwater storage can be calculated as the time integral of velocity at the surface.  Applying
Darcy's law at the surface (z=0) and integrating gives:

$$u = \Delta S = \int_0^t K\frac{dh}{dz}\bigg|_{z=0} dt' \tag{14}$$

Equation (14) can be computed by differentiating Eq. (11) w.r.t. $z$ and then numerically integrating over time.  Alternatively,
the change of storage can be reported from the numerical model.
**2.3 Numerical solution**
We used the COMSOL Multiphysics® software, validated against the analytical solutions for uniform permeability, to solve
the stress and flow equations (1) and (4) . The finite-element model is unrestricted in terms of spatial distribution of parameter
properties and in terms of the boundary condition functions.



### 2.4 Parameter allocation

Selected parameter values for the BAS underlying the GBM floodplains are given in Fig. 2. The bulk values for the uniform representations are close to the harmonic average of the series components. We next discuss the context in which these parameter selections are made.

### 2.4.1 Modulus of elasticity, storativity and loading efficiency

Text-book $S_s$ values (Domenico and Schwartz, 1998) for the materials in the Bengal Basin range between approximately $1\times10^{-5}$ m$^{-1}$ (dense sandy gravel) and $1\times10^{-2}$ m$^{-1}$ (plastic clay). In large-scale modelling of head recession data in the basin Michael & Voss (Michael and Voss, 2009a) achieved their best fits when $S_s$ was $9.4\times10^{-5}$ m$^{-1}$ taking pumped abstraction to be areally uniform. This is the basis for the range in specific storage, $S_s$, for the BAS (Fig. 2).

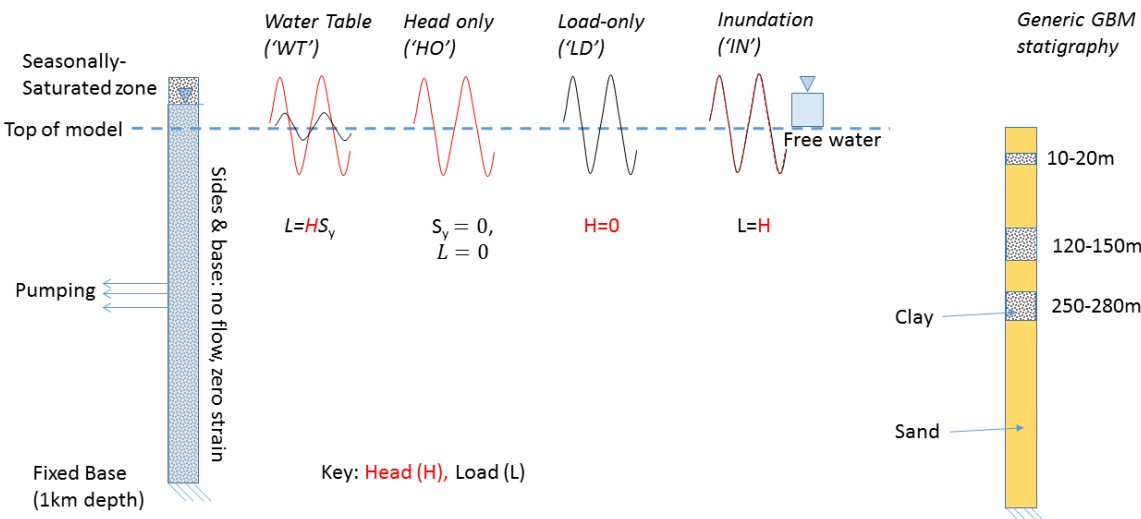

| | Uniform | Layered representation | | | | | | |
|---|---|---|---|---|---|---|---|---|
| | *homogeneous* | *1 (sand)* | *2 (silty-clay)* | *3 (sand)* | *4 (silty-clay)* | *5 (sand)* | *6 (silty-clay)* | *7 (sand)* |
| Thickness (m) | 1000 | 10 | 10 | 100 | 30 | 100 | 30 | 720 |
| $S_y$ (-) | 0.1 [1] | 0.1 | - | - | - | - | - | - |
| $S_s$ (m$^{-1}$) | 0.00001 [2] | 1 x 10$^{-5}$ | 1 x 10$^{-4}$ | 1 x 10$^{-5}$ | 1 x 10$^{-4}$ | 1 x 10$^{-5}$ | 1 x 10$^{-4}$ | 1 x 10$^{-5}$ |
| $K_v$ (ms$^{-1}$) | 0.00000005 [3] | 1 x 10$^{-5}$ | 1 x 10$^{-8}$ | 1 x 10$^{-5}$ | 1 x 10$^{-8}$ | 1 x 10$^{-5}$ | 1 x 10$^{-8}$ | 1 x 10$^{-5}$ |



| $E$ (MPa) | 82.07 | 850.89 | 82.07 | 850.89 | 82.07 | 850.89 | 82.07 | 850.89 |
| $\beta$ (-) | 0.996 | 0.961 | 0.996 | 0.961 | 0.996 | 0.961 | 0.996 | 0.961 |
| $\xi$ (-) | 0.993 | 0.932 | 0.993 | 0.932 | 0.993 | 0.932 | 0.993 | 0.932 |


**Figure 2. The 1D model showing (top) the upper surface boundary conditions with head as red lines and mechanical load (weight)**
**as black lines, expressed as metres of water; and a representative stratigraphy for the BAS underlying the GBM floodplains, with**
**the profile depth being 1 km; and (bottom) parameter values for the uniform and layered 1D representations. Porosity is taken as**
**0.1 throughout; $\nu=0.25$; $E$, $\beta$ and $\xi$ are calculated using Equations (5) and (6). [1] Shamsudduha et al., 2011; [2] Burgess et al., 2017; [3]**
**Michael and Voss, 2009b.**

Specific storage $S_s$ and Young's Modulus $E$ are related though Eq. [5] and to the loading efficiency $\beta$ via Eq. (6). These inter-
relationships are plotted in Fig. 3. It is notable that for $E<1$ GPa, $\beta>0.95$ and $S_s>1\times10^{-5}$ m$^{-1}$. Thus the loading efficiency only
falls significantly below 1 for materials stiffer than around 1 GPa, and where the specific storage is less than $1\times10^{-5}$ m$^{-1}$.
Uncemented sediment is thus expected to have $\beta\sim1$ (Bakker, 2016); on this basis the BAS sediment is unlikely to be
sufficiently stiff in the top few hundred metres to allow decoupling of the stress and flow equations. This is confirmed by in
situ, high-pressure dilatometer measurements(de Silva et al., 2010) giving $E$ within the broad range for sediments given in Fig.

192 3.





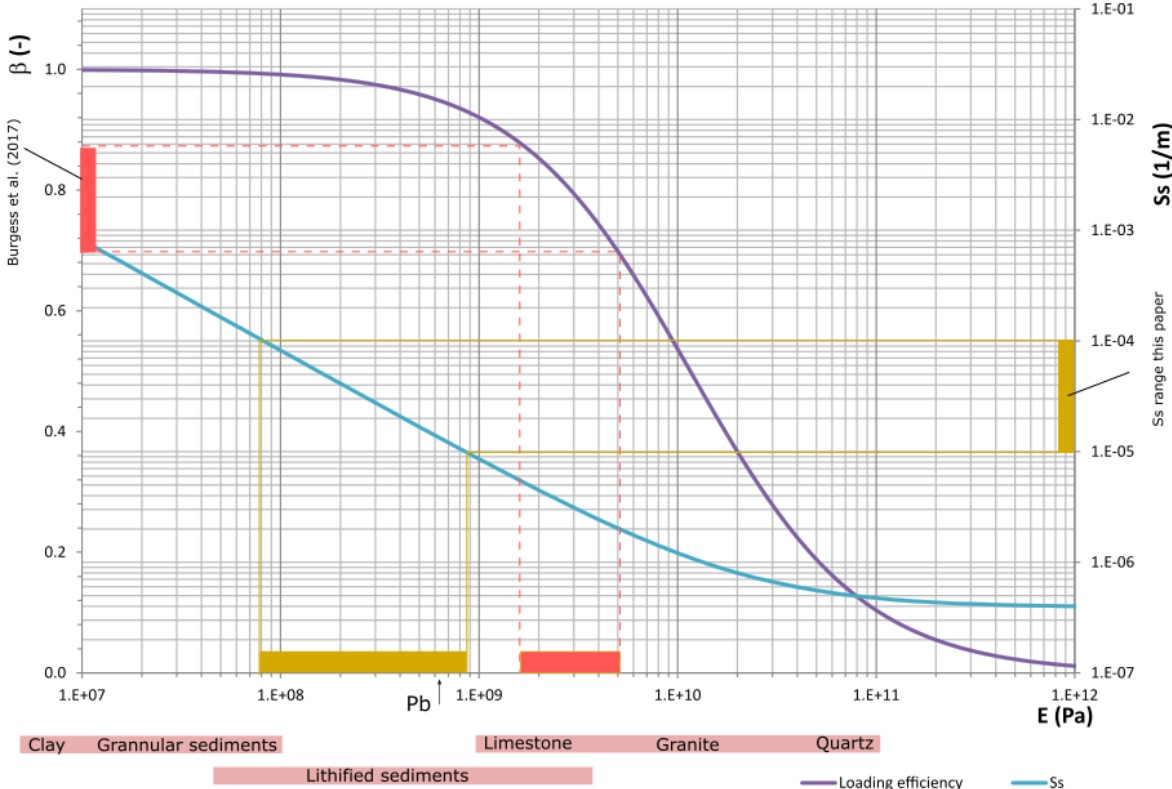


**Figure 3. Relationship between 1D Specific storage ($S_s$), Young's modulus ($E$) and 3D loading efficiency ($\beta$) using equations (5) and (6) assuming porosity of 0.1 and Poisson's ratio of 0.25. Projections show the corresponding inferred ranges of $E$ based on the $S_s$ range applied ($1 \times 10^{-5}$ - $1 \times 10^{-4}$ m$^{-1}$) and the loading efficiencies calculated via barometric efficiency estimates (0.69-0.87) by Burgess et al. 2017. Pink bars show indicative ranges for common geological materials. Arrow indicates data from 73 m depth at Padma Bridge (Pb) (De Silva *et al.*, 2010).**

Estimates of loading efficiency based(Jacob, 1940) on barometric efficiency are rather lower: a range of 0.69-0.87 has been determined at Laksmipur in the GBM sediment(Burgess et al., 2017). This is potentially indicative of a considerable stiffening due to burial ($E$ in the range 6-17 GPa), indicating $S_s$ in the range $1 \times 10^{-6}$ to $9 \times 10^{-8}$ m$^{-1}$. Such a condition might be expected in a Gibson soil (Gibson, 1974; Powrie, 2014). However, the Laksmipur estimates do not decrease systematically with depth, possibly due to changes in stiffness in different materials. Therefore for the purposes of this paper we adopt $S_s$ estimates based on field measurements and use the corresponding $\beta$ and $E$ values.

## 2.4.2 Hydraulic conductivity

Basin scale modelling suggests a horizontal-vertical anisotropy for hydraulic conductivity in the BAS of ~10,000 (Michael and Voss, 2009a; Ravenscroft et al., 2005). This may be explained as an effective, large-scale value incorporating finer-scale detail of the highly heterogeneous sedimentary record of the past deltaic environment where low permeability lenses and





drapes are laterally discontinuous (Hoque et al., 2017). Michael and Voss (2009b) cite aquifer tests (Hussain and Abdullah,
2001) conducted by the Bangladesh Water Development Board (BWDB) giving a range for hydraulic conductivity ($\kappa$) from
$3{\times}10^{-5}$ to $1{\times}10^{-3}$ m s$^{-1}$. Accounting for anisotropy, $\kappa_v$ may therefore locally be in the range ~$1{\times}10^{-9}$ to $1{\times}10^{-7}$ m s$^{-1}$. The $\kappa_v$
values of the uniform and layered representations of the BAS underlying the GBM floodplains (Fig. 2) and of silty-clay in
layered representations of the Khulna and Laksmipur sites (Sect. 4) lie within this range.

### 2.4.3 Specific yield

Specific yield is the drainable porosity of the material in which the water table moves. Michael and Voss (2009a) cite a range
from 0.02 to 0.19 in Bangladesh, noting that much of the Basin has a specific yield in the range of 0.02–0.05. We take $S_y$=0.1
and 0.01 as order-of-magnitude values typical for sand and clay respectively (Domenico and Schwartz, 1998).

### 2.5 Upper boundary conditions and groundwater abstraction

Changes to the shallow water budget which have the potential to be laterally-extensive and uniform include: water arriving as
rainfall at the surface and either ponding or moving to the shallow water table as recharge; and water departing the surface or
the water table by evaporation, or as runoff to the extensive network of drainage channels. Pumping for domestic and irrigation
supply may potentially be considered as areally-uniform, where sufficiently common and over a wide area (Michael and Voss,

223   2008).

The changing shallow water budget causes a change in mechanical loading to the aquifer system, and if in direct hydraulic
continuity with the saturated water column it also causes a change in head. If the shallow water is not hydraulically connected
to the saturated aquifer system, the effects of the changing water budget are transmitted to depth by mechanical
compression/extension of the sediment, but not by hydraulic diffusion.
Changes to the barometric pressure also apply a laterally-extensive changing force to the surface of the aquifer and to the water
column, and earth tides are also laterally-extensive. Both effects are neglected for simplicity here.
To explore the consequences of these hydraulic and mechanical loading sources, the groundwater dynamics associated with
three upper surface boundary conditions are modelled here (Fig. 2).
Firstly, the effect of a changing level of free water is examined, such as would be seen in paddy-fields, ponds or during
floodwater inundation. This condition is here termed 'IN'. The change in free-water level is equal to both the change in head
and the change in mechanical load at the upper surface (load is here parameterised in metres of water rather than as a stress).
Secondly, the effect of changes to unconfined storage due to a moving water table is examined. This condition is here termed
'WT'. The change in load is the specific yield times the head. For very small specific yields this condition approaches the
hydraulic-only ('HO') loading case, whereby there is insignificant mechanical load, despite the change in head.
Thirdly, we examine the effect of a changing surface water store (which could be either free water held above an impermeable
barrier, or a perched phreatic aquifer) which is hydraulically isolated from the main aquifer system. A mechanical load only is
applied, therefore no head change is applied to the aquifer and this condition is termed 'LD'.





These three TWS loading scenarios are applied in turn to a uniform and a layered representation of the BAS underlying the
GBM floodplains. The loading is applied as sinusoidal functions with unit amplitude and time period of 1 year to simulate the
annual hydrological cycle.
Additionally, the effects of groundwater abstraction are simulated. Abstraction is taken evenly from the depth interval 50-
100 m at an average rate of 0.2 m a$^{-1}$, either as continuous pumping or as discontinuous pumping $\pi$ out of phase with the TWS
load, as a coarse representation of seasonally-varying pumping for irrigation during the dry season.

## 3 Forward modelling results

The modelled responses of groundwater head to sinusoidal hydraulic and mechanical source terms, together with changes in
groundwater storage and ground surface vertical displacements, are illustrated for the GBM environment with uniform
properties in Figures 4 and 5. Figure 4 shows the modelled responses over ten years at depths of 30, 100 and 300 m,
approximating typical BWDB multi-level piezometers (BWDB, 2013). The depth variations of amplitude and phase for
groundwater head and the phase-lag for surface displacement are summarised in Fig. 5. The effect of layering (Supporting
Information) is to cause departure from the uniform cases, so interpretation of data in a real, heterogeneous aquifer should take
into account local deviation from idealised uniform conditions. However, in general, the loading style ('IN', 'WT', 'LD') and
pumping regime are of more significance for the head responses and surface displacements than the detail of the BAS
stratigraphy.

### 3.1 The free surface water inundation scenario ('IN')

Under free-surface water inundation, head changes are characteristically equal in amplitude at all depths and in-phase with the
inundation signal. Away from the top boundary, the instantaneous head due to loading in this case is $h = \xi L$. Since $\xi$ is close
to 1 and $H = L$, the head is everywhere almost equal to the mechanical load given that at the top boundary the head is also
$h = H$. Therefore under free-surface water inundation in the absence of pumping, piezometers at all depths can be expected to
record the surface water mechanical load, effectively operating as weighing lysimeters. The vertical displacement of the ground
surface is extremely small (amplitude ~0.4 mm), being due to the small compression of porewater itself over the 1 km
simulated depth, and is out of phase with the load (i.e. the ground surface moves downwards under an increasing load). The
amplitude of change in saturated storage is infinitesimal (~0.02 mm). The system is essentially 'un-drained'; water does not
flow in or out of the pores which therefore experience only minimal strain.





| Scenario | Head (m) | Storage (dashed line) and displacement (solid line) (m) |
|---|---|---|
| (a) 'IN' | | |
| (b) 'WT' $S_y$=0.1 | | |
| (c) 'WT' $S_y$=0.01 | | |
| (d) 'LD' | | |
| (e) 'IN' Const. Q | | |





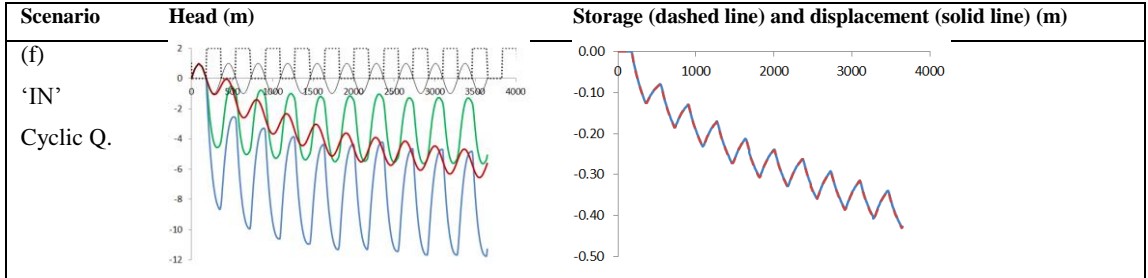

**Figure 4. 1D model simulations for the GBM environment, showing results for the scenarios (a) 'IN', (b) 'WT' ($S_y$=0.1), (c) 'WT' ($S_y$=0.01), (d) 'LD', (e) 'IN' with constant pumping, (f) 'IN' with cyclic pumping, (see text for explanation). The x-axis is time in days, shown to 10 years (i.e. 3650 days). The amplitudes reported in the text are calculated from the max-min of the last annual cycle. Left: The y-axis is head, in metres (m). The surface head and/or mechanical load boundary conditions (black line) are expressed as equivalent m head (for the WT condition the unit variation of head is given and the $S_y$ variation in mechanical load is not shown); results are in green (30 m depth), blue (100 m depth) and red (300 m depth) in all cases. For (a) results are co-linear at all depths; for (f) the intermittent pumping is shown as off/on by the square-wave dotted line. Right: The y-axis has dimension of length, in metres (m), showing changes in storage (dashed red line) and surface displacement (solid blue line) for each scenario.**

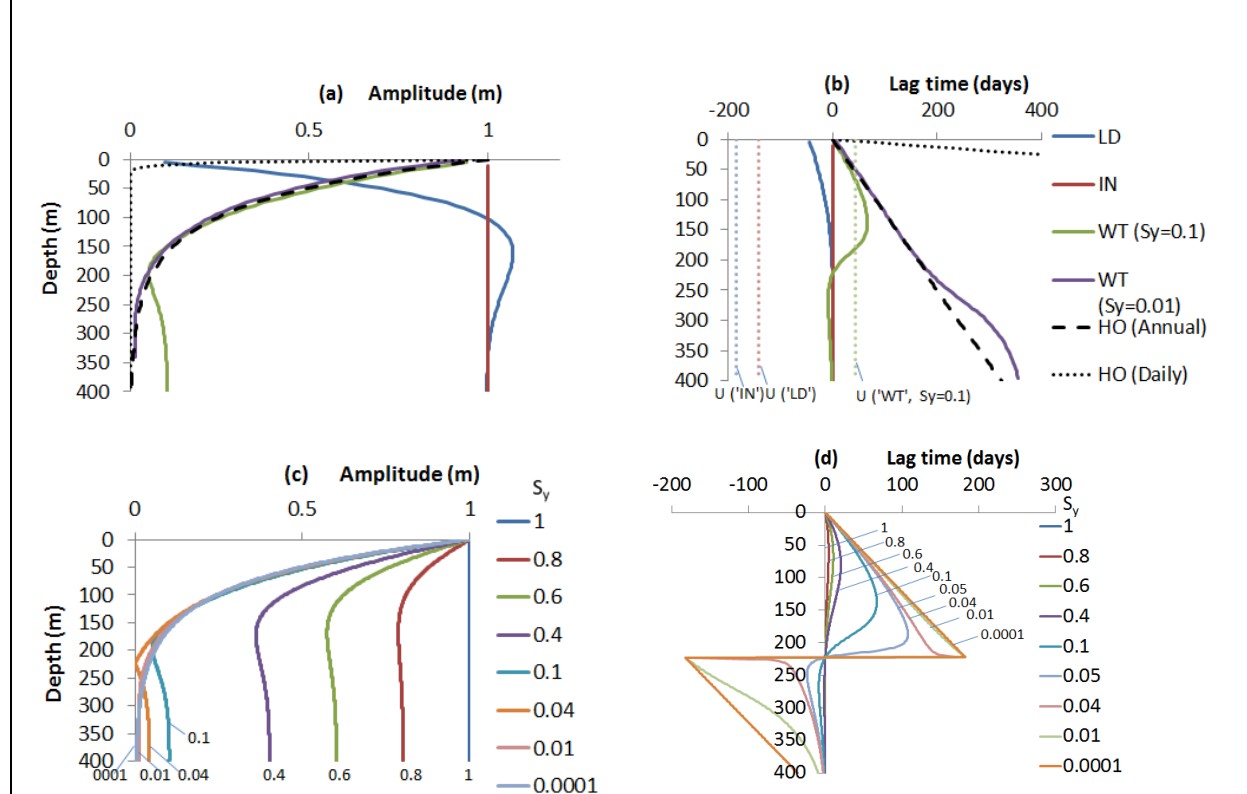

**Figure 5. Profiles with depth for (a) amplitude of head response, (b) phase of head response and surface displacement (U), (c) sensitivity of amplitude to $S_y$ for the 'WT' boundary condition, (d) sensitivity of phase to $S_y$ for the 'WT' boundary condition. For (a) and (b) the colour code for the scenarios 'LD', 'IN', 'WT' ($S_y$=0.1), 'WT' ($S_y$=0.01), and HO, is shown in the top right panel (see text for explanation); in (b), displacement for the WT, $S_y$=0.01 scenario overlies that for the WT, $S_y$=0.1 scenario, so is not shown.**



## 3.2 The variable water table scenario ('WT')

By contrast with the 'IN' scenario, head changes determined by a moving water table are depth-variable in amplitude and phase. When $S_y \rightarrow 0$ the 'WT' condition tends to the head-only end-member ('HO') and when $S_y \rightarrow 1$ the 'WT' condition tends to the 'IN' scenario. The maximum lag for $S_y = 0.1$ is at 137 m depth (or $\theta = 1.94$ ), beyond which it reduces (Fig. 5b). The sensitivity in head to $S_y$ for the 'WT' scenario is illustrated in Fig. 5c. The amplitude of head responses is less than the water table fluctuation at all depths. Moreover, only a deep piezometer such as the one indicated at 300 m (Fig. 4b) will behave as a weighing lysimeter in this scenario. Here, heads are in phase with the water table and have approximate magnitude, $h = \xi L = \xi S_y H$, as in the study by van der Kamp and Maathuis (1991) of a thick aquitard overlying a confined aquifer. At 100 m the amplitude of head change is greater than at 300 m, and lags behind the water table. At 30 m the amplitude of head change is greatest and the lag is less than at 100 m. The difference in the head responses compared to the 'IN' scenario is due to the difference in magnitudes of the applied head and applied load under the 'WT' scenario, causing an instantaneous internal head gradient which subsequently diffuses. Ground surface displacement is ~4 mm and lags the load by 44 days. With increased head at the top boundary, the upper surface moves upwards because as higher heads penetrate the aquifer the effective stress is reduced. The lag is due to the time taken for the surface head to diffuse downwards.

## 3.3 The hydraulically disconnected load scenario ('LD')

Heads in the case of a surface load hydraulically isolated from the aquifer show a third characteristic behaviour. In this case the amplitude of head change increases from zero at the top boundary (Fig. 5a), and counter-intuitively reaches a peak which is greater than the load, 1.07 m at 162 m (or $\theta$ =2.29). The amplitude thereafter tends to $\xi L$ at greater depth, whilst the lag tends to zero. Therefore heads in relatively deep piezometers potentially represent the surface load under a 'LD' boundary condition, as in Fig. 4d where the heads at 300 m match the surface load, whereas at 30 m they do not. This is due to upward head diffusion towards the surface where the head boundary condition is $h$=0. The lag which occurs in the 'WT' scenario due to the applied head exceeding the mechanical load is reversed in this 'LD' scenario, becoming a lead time as the applied load exceeds the applied head. Surface displacement is out of phase with the load, leading by ~π radians. The ground surface displacement amplitude of ~4 mm is ten times greater than for the 'IN' scenario but is still very small in comparison to the annual variability of order 10 cm measured by GPS(Steckler et al., 2010).

## 3.4 The influence of pumping

Introduction of pumping from the depth interval 50-100 m causes hydraulic dis-equilibrium which continues well beyond the ten years' simulation, as the head drawdown propagates deep into the profile. As well as drawing water from storage at depth, pumping induces recharge from the surface, there being a downward hydraulic gradient from the surface to the pumped horizon, and upwards from the deeper levels to the pumped horizon. Variable perturbation due to the 'IN' surface load is nevertheless clearly evident in the deep groundwater head measurements following correction for secular decline (Fig. 4e).





Elastic displacement, manifested as ground surface decline, exceeds 40 cm after ten years of pumping but, as in the un-pumped
'IN' scenario, the annual fluctuation due to surface loading is vanishingly small (0.03 mm). Thus, in addition to the possibility
of irreversible plastic deformation, elastic strain may gradually increase due to continuous pumping as stored water is drawn
from increasing depths.
Intermittent pumping strongly increases the seasonal variation in heads at the depth of pumping and this disturbance diffuses
to adjacent levels. However, as in the case of continuous pumping, the surface load signal is largely preserved in the deep
groundwater head response at 300 m. Also, intermittent pumping induces the same average long-term secular decline in stored
water volume and ground surface displacement as continuous pumping, but with additional annual fluctuation caused by the
pump switching on and off (decline/drawdown during the dry period when the pumps are used for irrigation and recovery
during the rainy season when the pumps are off).
**3.5 Model results for ground surface displacement**
Taking into account a small correction for the compressibility of water, surface displacement in the model is almost equal to
the total change in elastic storage in the permanently saturated aquifer. For the cases where pumping dominates the removal
of water, surface displacement is in phase with the pumping (Fig. 4f). For the cases which set up a diffusion of the hydraulic
signal between the surface boundary and the aquifer, the phase of surface displacement depends on the hydraulic (non-loading)
head changes at all depths (Fig. 4b,c,d). Therefore the lag for vertical displacements under the 'LD' surface condition is ~π
out of phase with displacement under the 'WT' condition. Note from Eq. [12] that the amplitude and lag are both a function
of $\theta = z\sqrt{\dfrac{\omega}{2D}} = z\sqrt{\dfrac{\pi}{DT}}$ and therefore the solutions given here would be scaled in $z$ by any changes to bulk diffusivity, $D$, and
signal frequency (or time period, T): higher frequency would give the same distribution but for a smaller $z$ and the reverse
would be true for diffusivity. Intermittent pumping produces the largest cyclic displacements, however, in the order of
centimetres, because this condition causes the greatest volume of seasonal drainage from the formation itself. Where there is
non-uniform loading, as produced for example by a variable river stage, lateral groundwater drainage may occur and surface
vertical displacements may be greater under these conditions too.

**4 Applying the partial coupling analysis to field data**
Applying the 1D partial-coupling analysis to field data, we examine poromechanical perturbations at two sites, Khulna and
Laksmipur and in southern Bangladesh (Fig. 1). Hourly measurements of groundwater pressure made between April 2013 and
June 2014 in three closely-spaced piezometers between 60 and 275 m depth at each site are illustrated as hydrographs of
equivalent freshwater head in Supporting Information. The objective here is to apply the principles and assumptions of the
partially-coupled hydro-mechanical approach to reproduce the characteristic features of the multi-level groundwater



hydrographs using broadly representative aquifer parameters, rather than to attempt an exact match by inverse modelling.
Inspection of the hydrographs at both sites indicates, by reference to Figures 4 and 5, that mechanical loading significantly
influences the measured heads. Additionally, the presence of thick clay aquitards at both sites (Figures 6, 7) suggests conditions
under which heads may be determined solely by mechanical loads and piezometers might behave as geological weighing
lysimeters; a possibility which we put to the test.
The approach at each site is as follows:
i. A two-component sand-clay stratigraphy is based on site data, and parameter values are selected from the ranges described
in Section 2.
ii. The piezometric readings are compared to examine possible pumping influences which need to be taken into account in the
model by means of a simple abstraction pattern. Based on what is known about nearby abstractions an appropriate pumping
depth interval is determined. The magnitude of the extraction rate is manually adjusted as a fitting parameter.
iii. Where a piezometer is uninfluenced by pumping we test its behaviour as a geological weighing lysimeter. The heads in the
chosen piezometer are assumed to define the mechanical load at the surface, and this assumption is tested for self-consistency
by comparison of the simulations to the data from all three piezometers.
iv. The nature of the upper head boundary is then examined by reference to the implications for a variety of hydraulic loading
conditions.  For a 'WT' boundary, changing $S_y$ manually as a fitting parameter adjusts the magnitude of the applied heads
concomitant with the mechanical load.
**4.1 Groundwater levels at Khulna, south-west Bangladesh**
At Khulna town(Burgess et al., 2014) piezometers KhPZ60, KhPZ164 and KhP271 (the numbers indicate depth to the
piezometer screen in metres) are located 700 m from the ~300 m wide tidal Rupsa River, in a grassy compound which also
contains municipal water-supply pumping boreholes (Supporting Information). The lithological sequence (Fig. 6) comprises a
surface clay layer overlying sand in which KhPZ60 is screened, and a deeper layer of clay at 100 m separating the shallow
sand from a deeper sand formation in which KhPZ164 and KhPZ271 are screened. Year-round pumping from 250-300 m depth
maintains a consistent downward head difference of ~3 m between the uppermost and the lower two piezometers. It is the
transient head variations rather than the absolute steady-state head differences that are of interest here. Bodies of standing
water in the vicinity, water in the unsaturated zone, and shallow groundwater combine with the sinuous Rupsa River as sources
of TWS load; groundwater pumping is an additional source of hydraulic variation.
The three Khulna hydrographs are characterised by periodic variations containing tidal frequency components throughout the
rising and falling limb of the annual cycle, and a series of episodic increments superimposed on the rising limb during the
monsoon season; the annual amplitude of groundwater head variation is ~2.5 m. Amplitude of the tidal frequency components
increases between 60 m and 164 to 271 m depth, with no phase lag and with a consistent synchroneity between the piezometer
heads and the Rupsa River water level fluctuations including the semi-diurnal and spring-neap cycles (Fig. 6 and Supporting
Information). Episodic deflections on the hydrograph rising limbs, coincident with rainfall events, are likewise simultaneous



at all measurement depths (Burgess et al., 2014). Therefore by reference to the partial coupling analysis (Figures 4 and 5) It is
evident that heads in the Khulna piezometers respond primarily to mechanical loading by a combination of monsoon water
and tidal loading.
At a daily level the time series of groundwater heads in KhPZ164 and KhPZ271 include an additional frequency component
which simple analysis of head differences confirms as the hydraulic influence of the daily municipal pumping schedule from
which KhPZ60 is protected by an intermediate clay layer. Therefore KhPZ60 alone is taken as recording a solely mechanical
loading response and the KhPZ60 head record is applied as the upper boundary condition to represent the varying TWS load
at the surface in a 1D hydro-mechanical model of the Khulna site (Fig. 6), assuming $\beta=1$. The upper boundary resolves all
sources of load acting at the site including from the Rupsa River, which is a linear rather than an areally-extensive load. The
ratio of daily variability in head at KhPZ60 and in the Rupsa River level is ~0.06, therefore the 1.23 m annual variation in river
stage would explain ~0.07 m head variation in KhPZ60, only 3% of the total. Therefore 97% of the annual variation in head
at KhPZ60 is attributable to changes in TWS other than load transmitted from the river, representing areally-extensive loads
as required by the 1D partially-coupled analysis. Given the relatively well-drained urban context at Khulna and the absence of
areally-extensive open water that otherwise characterises the rural areas of the GBM floodplains, a 'WT' condition is most
likely the dominant loading style, but other sources of loading may also contribute. The layered structure of the Khulna model
(Fig. 6) has clay at 0-50 m and 100-150 m with sand in between. The daily municipal pumping cycle is implemented as a
source term of 2.4 m a$^{-1}$ for 12 hours of each day applied over the interval 200 to 350 m, the rate having been manually adjusted
by reference to the daily head fluctuations in KhPZ164 and KhPZ271.






















**Figure 6. Khulna: comparison of observed heads (solid lines) and simulated heads (dashed lines), starting 27 April 2013, for WT upper boundary condition ($S_y$=0.4). X-axis is time in days. The surface loading is set equal to the observed head in KhPZ60, and the surface head is set to the observed head in KhPZ60 divided by $S_y$. The pumping rate is 2.4 m a$^{-1}$ for 12 hours of each day, switching on at 05:45 am. Top (green) is KhPZ60, middle (blue) is KhPZ164, bottom (red) is KhPZ271.**




Figure 6 compares the measured groundwater heads with the heads simulated by the model under the assumption of a 'WT'
boundary with $S_y$ assigned a value of 0.4, with $\kappa_{sand} = 1 \times 10^{-5}$ m s$^{-1}$, $\kappa_{clay} = 1 \times 10^{-9}$ m s$^{-1}$, $S_S = 10^{-4}$ m$^{-1}$ (corresponding to
$E$=82.07 MPa), $\nu = 0.25$ and $n = 0.1$. The results are insensitive to $S_y$ being varied in the range from 0.1 to 1 (the latter being
equivalent to an 'IN' boundary), and are near-identical in the case of a 'LD' boundary (Supporting Information). This is
because the upper clay effectively isolates the piezometers from the surface hydraulically.

### 4.2 Groundwater levels at Laksmipur, south-west Bangladesh

At Laksmipur(Burgess et al., 2017) the piezometers LkPZ91, LkPZ152 and LkPZ244 are situated in a rural region of rice-
paddy and tree plantations on the Lower Meghna floodplain (Supporting Information), 10 km distant from the River Meghna
and 8 km from municipal boreholes which pump from 270–300 m depth. Seasonal pumping from depths up to 100 m for rice
irrigation is common in the vicinity. The lithological sequence indicates fine sand with occasional silty clay layers. The
hydrographs are characterised by a sequence of episodic increments in groundwater head associated with periods of heavy
rainfall producing a rising limb of amplitude ~1 m through the monsoon season; during the dry-season recession, minor
periodic fluctuations of order 0.01 m containing atmospheric frequency components become more clearly evident(Burgess et
al., 2017). The episodic increments are almost synchronous and of consistent magnitude at all piezometer depths, indicative
by reference to Figures 4 and 5 of groundwater heads responding dominantly to mechanical loading and unloading due to
changes in TWS above the aquifer surface.
Here, cyclical head differences between LkPZ244 and the shallower two piezometers   indicate hydraulic influences of dry-
season pumping on the LkPZ91 and LkPZ152 hydrographs, whereas downward propagation of the hydraulic signals to
LkPZ244 is prevented by the clay layer between 170 and 200 m depth. Therefore LkPZ244 is taken as recording a solely
mechanical loading response and the LkPZ244 head record is applied as the upper boundary condition to represent the varying
TWS load at the surface in a 1D hydro-mechanical model of the Laksmipur site (Fig. 7). All styles of upper boundary were
applied ('IN', 'LD', and 'WT' with a range of $S_y$ values, see Supporting Information D) in an attempt to distinguish the
dominant source of TWS load around the site from the boundary style leading to the best fit with piezometer measurements.
In all other respects the models incorporate the dimensions and assumptions as described in Sect. 3, with sand ($\kappa_{sand} = 1 \times 10^{-5}$ m s$^{-1}$) and three clay layers(BWDB, 2013) at 25-30 m, 115-130 m and 170-200 m ($\kappa_{clay} = 1 \times 10^{-8}$ m s$^{-1}$), and E=82.07 MPa.
A simple dry-season pumping regime over a 105 day period starting 17 November 2013 is implemented as a source term of
0.04 m a$^{-1}$ applied over the interval 30 to 70 m in the model, manually adjusted by reference to the LkPZ91 and LkPZ152
hydrographs.





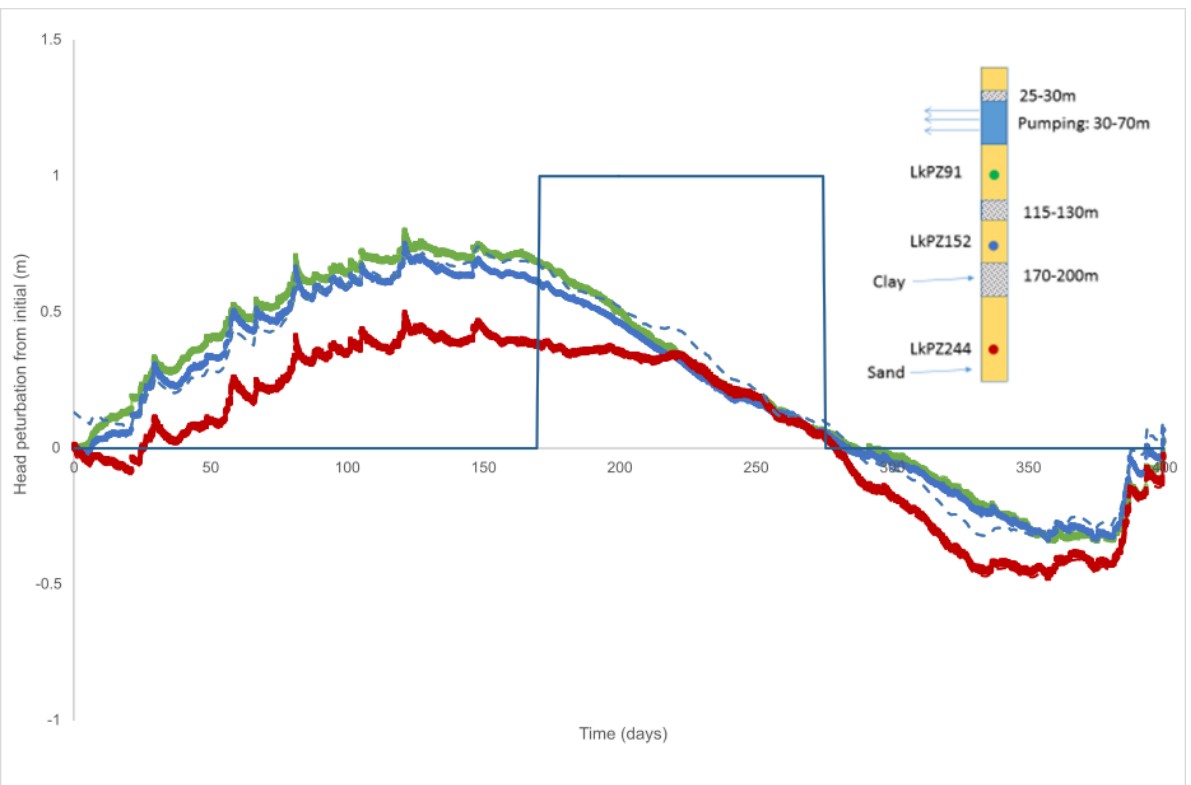


**Figure 7. Laksmipur: comparison of observed heads (solid lines) and simulated heads (dashed lines) starting 31 May 2013, for 'WT' upper boundary condition ($S_y$=0.8), for LkPZ91 (green), LkPZ152 (blue) and LkPZ244 (red). X axis is time in days. The surface loading is set equal to the observed head in LkPZ244, and the surface head is set to the observed head in LkPZ244 divided by $S_y$. The pumping rate is 0.04 m a$^{-1}$ for the period shown (1 for 'on', 0 for 'off').**


For LkPZ244 the simulated heads are an excellent match with measurements over the entire period. The simulated heads for
the shallower two piezometers LkPZ91 and LkPZ152 most closely match the measurements under a 'WT' boundary with $S_y$
assigned a value of 0.8 (Fig. 7 and Supplementary Information). The consistently higher simulated heads compared to
observations at LkPZ152 could be simply explained by the sands at that depth having a lower loading efficiency. The model
results therefore confirm that LkPZ244 is isolated from the hydraulic effects of water table variation and of seasonal pumping,
and the LkPZ244 groundwater head variation over the observation period is determined solely by mechanical loads at the
surface. Therefore LkPZ244 is validated as acting effectively as a geological weighing lysimeter (Burgess et al., 2017).
For the shallower piezometers, the best fit value for $S_y$ is higher than is reasonable for fine sand and more likely indicates the
combined effects of a variable water table and fluctuating levels of standing water, in drainage channels and on paddy fields
around the piezometer site, consistent with the field situation. As a consequence of seasonal pumping at 0.04 m a$^{-1}$, the model
shows groundwater is both drawn from storage and induced as recharge from the upper surface, but the amplitude of saturated





storage fluctuation is only 6 mm, therefore changes to the water budget are dominated by recharge to the water-table. The
surface displacement is predicted at 6 mm amplitude, in phase with the changes in storage.
**5 Discussion**
**5.1 Aquifer responses to discrete modes of terrestrial water variation**
Models based on the 1D partially-coupled hydro-mechanical analysis confirm that substantial poroelastic influences should be
expected in the Bengal Aquifer System, and that groundwater heads respond characteristically to changes in specific terrestrial
water stores (Figures 4 and 5). Only laterally-extensive flooding above an aquifer fully saturated to the ground surface (the
'IN' loading style) will drive instantaneous and synchronous head variations at all depths determined by the loading efficiency,
inducing negligible flow of groundwater. In any situation involving a variable water table (the 'WT' loading style) and for any
variable loads hydraulically disconnected from the aquifer (the 'LD' style), hydraulic gradients are imposed due to the unequal
magnitude of stress and head at the surface. These gradients take time to dissipate, depending on the frequency of the signal
fluctuation and the aquifer hydraulic diffusivity, and so lead to differences in amplitude and phase of the head response with
depth. In these situations, the relative importance of the hydraulic and mechanical influence is controlled by the aquifer
hydraulic diffusivity, the loading efficiency and the depth of interest. In the case of a fluctuating water table, the difference
between the head and stress signals is a function of the specific yield, $S_y$, in the zone of fluctuation.
The characteristic responses of the aquifer might therefore provide a key to identifying the terrestrial water store dominating
ΔTWS, by monitoring vertical profiles of groundwater head. Multiple terrestrial water stores will normally contribute,
however, as at Laksmipur and Khulna, so a unique identification may not be possible. This limitation is inherent to the 1D
analysis, which resolves all the contributions to load into one upper boundary condition respectively for head and stress. The
analysis indicates how different loads and dynamic responses superpose to produce the observed groundwater hydrographs. In
principle, key aspects of the water balance may be better estimated by de-convolving known components of the ΔTWS signal.
At Khulna and Laksmipur, the 1D partially-coupled analysis leads to good agreement between simulated and observed heads
consistent with the local conditions, confirming it as a suitable basis for representing the poroelastic behaviour of the BAS.
**5.2 Significance for groundwater monitoring and geological weighing lysimetry**
In terms of the extent to which piezometer water levels indicate recharge and drainage, it is only where there is a rapid hydraulic
connection between the piezometer and the water table that the piezometer will be sensitive to head change at the water table
and therefore to changes in unconfined storage. If a piezometer is hydraulically isolated from surface water and/or the water
table and is beyond other transient hydraulic influences, it can respond to changes in the weight of the TWS load, acting as a
geological weighing lysimeter (Smith et al., 2017; van der Kamp and Maathuis, 1991). In this case, where the changing load
is due to a moving water table, knowledge of the loading efficiency allows the load measurement to be converted into an
estimate of recharge and discharge.



In all other situations, a wide range of coupled hydro-mechanical responses can be expected, as we have shown for the BAS
(Figures 4 and 5). Seasonally-variable groundwater heads (Fig. 4) are therefore open to misinterpretation as seasonally-variable
groundwater storage, leading to error in determination of recharge if the poroelastic nature of the response is neglected.
Consider heads at 30 m, a common depth for Bangladesh Water Development Board (BWDB) monitoring boreholes
(Shamsudduha et al., 2011). For the case of a variable load hydraulically disconnected from the aquifer (Fig. 4d) the annual
water level rise is equal to half the amplitude of the load yet augmentation of elastic storage, by definition in this case, is nil.
For the case of variable TWS inundation (Fig. 4a) the annual groundwater level rise is equivalent to the annual depth of
inundation yet augmentation of elastic and unconfined storage is insignificant. Conversely, relative to a variable water table
(Fig. 4b,c) groundwater fluctuation at 30 m depth is attenuated. Failure to account for this would lead to an underestimate of
recharge to unconfined storage by about 30%. The error increases as hydraulic diffusivity decreases, therefore errors could be
expected to be greater in the coastal regions of the Bengal Basin where the thickness of silty-clays is greater(Mukherjee et al.,
2007).  Considerable caution is therefore necessary in the use of even relatively shallow piezometers as indicators of recharge
to the water table. A true indication of recharge requires either a shallow tubewell screened over the depth interval of actual
water table fluctuation, or a deep piezometer responding as a geological weighing lysimeter to the varying mass provided by
a fluctuating water table. In the latter case it is recharge to the shallow water table that is measured, not recharge at the depth
of the piezometer.
The 1D hydro-mechanical framework can be applied as a test for the special cases where groundwater head responds solely to
mechanical load, and hence to validate the use of geological weighing lysimetry. The laterally-extensive loading criterion
inherent to the 1D analysis must apply, and the piezometer screen must be isolated or distant from hydraulic transients
originating at the surface or from pumping. We have shown for the BAS that these requirements most likely occur at depths
beyond about 250 m, as in the case of 'WT' and 'LD' loading styles in the absence of pumping (Fig. 5). The inundation ('IN')
style of TWS variation leads to instantaneous transmission of head without loss of amplitude at all depths; in this case
piezometers at all depths provide a mechanical record of ΔTWS rather than a hydraulic record of storage variation and to infer
recharge would lead to 100% error. Our analysis demonstrates a solely mechanical loading response at 244 m depth at
Laksmipur, below the level of seasonal irrigation pumping, and at 60 m depth at Khulna, above the level of deep pumping for
municipal water supply.
**5.3 Significance for ground surface displacements and groundwater storage changes**
The models also demonstrate the amplitude and phase of ground surface displacement as a hydro-mechanical consequence of
varying terrestrial water stores, and the significance of pumping (Fig. 4e and 4f).
Under simplifications associated with the 1D model, vertical surface displacements relative to a fixed model base at 1 km
depth are approximately equal to the change in elastic storage, the small difference being due to compressibility of water.
These changes are minor in the BAS under all TWS loading styles, in the order of mm, compared to the displacements in the
case of seasonal groundwater pumping which are in the order of cm. Seasonal surface displacements in the order of cm have



also been attributed to strain acting over a depth scale of hundreds of kilometres due to the load applied by monsoonal inundation over the entire Bengal Basin (Steckler et al., 2010). Strains due to seasonal groundwater pumping at shallow depths may therefore be in the same order of magnitude but out of phase with crustal stain, making ground surface deflections a poor proxy for changing elastic storage in the aquifer. As a corollary, interpretation of seasonal ground surface fluctuations across the GBM floodplains solely in terms of deep crustal deformation(Steckler et al., 2010) potentially requires reassessment in the light of BAS aquifer poroelasticity.

## 5.4 Limitations and further consequences

In our analysis we have based values for the 3D loading efficiency, $\beta$ (0.961-0.996) and Young's Modulus, E (82-851 MPa) in the BAS on field measurements of $S_s$, for the sake of internal hydro-mechanical consistency, but we have noted a discrepancy with lower values for the 1D loading efficiency $\xi$ (0.69-0.87) derived from determinations of barometric efficiency(Burgess et al., 2017). These differences require attention, but the overall conclusions on the significance of poroelastic behaviour in the BAS and the pattern of poroelastic responses characteristic of specific upper surface TWS boundary conditions are unaffected.

Under certain circumstances the extensive load assumption inherent in the 1D analysis may break down. Rivers, as linear sources of head and load, can be accommodated within the 1D framework where their contribution to the TWS load is minor as demonstrated at Khulna. In general however, rivers should be expected to impose laterally variable heads and require a more generalised 2D or 3D fully-coupled poro-mechanical treatment(Boutt, 2010; Pacheco and Fallico, 2015). An equivalent constraint applies to strains, an additional reason for surface displacement not to offer a secure proxy for groundwater storage in the BAS. The dense distribution of rivers, distributaries and drainage channels in the Bengal Basin makes the BAS widely vulnerable to loading effects that may not adequately be reduced to a 1D description; 13% and 47% of 1035 piezometers in the BWDB groundwater monitoring network lie within 1 and 5 km respectively of a river.

## 6 Conclusions

We argue that a 1D *partially-coupled* approach to hydro-mechanical processes, whereby the loading term is included in the flow equation without the need to simultaneously compute the elastic equation, is a suitable basis for representing the poroelastic behaviour of the Bengal Aquifer System when surface conditions can be treated as areally-extensive. Applying a 1D *partially-coupled* hydro-mechanical analysis we have shown how the BAS responds characteristically to specific sources of terrestrial water storage variation. Rivers can be incorporated as a component of the 1D load where their contribution is small, but in general will require a 2D or fully 3D treatment.

Groundwater levels, groundwater recharge, vertical groundwater flow and ground surface elevations are all influenced by the poroelastic behaviour of the BAS. Our results expose the error of the conventional assumption of de-coupled hydraulic behaviour which underlies previous assessments of recharge to the BAS. Also they demonstrate the complexities in applying





ground surface displacements as a proxy measure for variations in groundwater storage. We propose that the 1D *partially-*
*coupled* analysis can be applied to validate when geological weighing lysimetry is applicable in the BAS. In some situations,
geological weighing lysimetry offers an alternative approach to recharge assessment.

**Author contributions**

WGB conceived the study; NDW led the mathematical analysis and the numerical modelling; all authors contributed to the
scenario descriptions and consideration of the modelling results; NDW and WGB drafted the manuscript; all authors reviewed
the manuscript.

**Acknowledgments**

We acknowledge funding from the UK EPSRC Global Challenges Research Fund (UCL/BEAMS EPSRC GCRF award
172313) to WGB for research on *Poroelasticity in the Bengal Aquifer System and groundwater resources monitoring in*
*Bangladesh.* NDW thanks the University of Southampton for leave of absence during the course of the project. Field
measurements at Khulna and Laksmipur were made with the kind assistance of the Bangladesh Water Development Board
(BWDB) and financial support from the UK Department for International Development (DfID) under the project *Groundwater*
*Resources in the Indo-Gangetic Basin* (Grant 202125-108) managed by Professor Alan MacDonald of the British Geological
Survey. We are grateful to Professors Mike Steckler, Columbia University, and Humayun Akhter, Dhaka University, for useful
discussions on ground surface vertical motions in the Bengal Basin, and to John Barker and William Powrie for helpful
discussion of the fundamental processes at the start of the research. Dr. Mohammed Shamsudduha, University College
London, and Ms. Sarmin Sultana, Dhaka University are thanked for discussions on the hydrological context of the groundwater
level monitoring piezometers. The data used are listed in the references and illustrated in the Supplementary Information.

**Nomenclature**

$\alpha$         Proportion of mechanical load as head
$\alpha_B$       Biot-Willis coefficient, $1 - K/K_s$
$\beta$, C     3D loading efficiency, Skempton's coefficient, or 'tidal efficiency'
$\delta_{ij}$       Kronecker delta
$\varepsilon_{ij}$       Strain
$\theta$        $z\sqrt{\dfrac{\omega}{2D}} = z\sqrt{\dfrac{\pi}{DT}}$



| 580 | $\lambda$ | $2\alpha_B(1-2\nu)/3(1-\nu)$ |
|-----|-----------|------------------------------|
| 581 | $\nu$ | Poisson's ratio |
| 582 | $\xi$ | 1D loading efficiency |
| 583 | $\kappa$ | Hydraulic conductivity |
| 584 | $\rho$ | Water density |
| 585 | $\sigma_{ij}$ | Stress tensor |
| 586 | $\sigma_t$ | Total stress |
| 587 | $\psi$ | Lag (radians) |
| 588 | $\omega$ | Angular frequency |
| 589 | | |
| 590 | $a$ | River half-width |
| 591 | $B$ | Barometric efficiency |
| 592 | $E$ | Young's Modulus |
| 593 | $D$ | Hydraulic diffusivity |
| 594 | $g$ | Acceleration due to gravity |
| 595 | $G$ | Shear Modulus |
| 596 | $h$ | Head |
| 597 | $H(t)$ | Top boundary head |
| 598 | $H_0$ | Amplitude of top boundary head |
| 599 | $J$ | Fluid source term |
| 600 | $K$ | Bulk Modulus of porous medium |
| 601 | $K_f$ | Bulk modulus of the water |
| 602 | $K_s$ | Bulk modulus of the solid grains |
| 603 | $L(t)$ | Top boundary load |
| 604 | $L_0$ | Amplitude of top boundary load |
| 605 | $n$ | Porosity |
| 606 | $p$ | Pore pressure |
| 607 | $S_y$ | Specific Yield |
| 608 | $S_s$ | Specific storage |
| 609 | $S_{s3}$ | 3D Specific storage |
| 610 | $t$ | Time |
| 611 | $u$ | Vertical displacement |
| 612 | $x$ | Perpendicular distance from a river |



z        Vertical coordinate

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
