# Peer review of "A partially-coupled hydro-mechanical analysis of the Bengal Aquifer System under hydrological loading"

_Hydrology and Earth System Sciences, 2018_

## Referee Comment (RC1) · Anonymous Referee #1 · 24 Aug 2018

This paper is about a partially-coupled hydromechanical modeling of an aquifer. Authors have used a simple coupling scheme for this purpose. In terms of novelty, I don't see any new method or development. The only new thing I can see here is just the application, which I think is not enough. Besides, due to using the commercial software, COMSOL, the results do not seem too realistic. I expected to see some 3D models representing the deformation because of the loading. Therefore, this paper is not a journal paper and I don't recommend its publication.

---

## Author Comment (AC1) · 4 Sep 2018

Our response to Referee#1's comments take the four points in turn: POINT 1: "I don't see any new method or development. The only new thing I can see here is just the application, which I think is not enough." POINT 2: "Due to using the commercial software, COMSOL, the results do not seem too realistic." POINT 3: "I expected to see some 3D models representing the deformation because of the loading." POINT 4: "Therefore, this paper is not a journal paper."

POINT 1: "I don't see any new method or development. The only new thing I can see here is just the application, which I think is not enough" In the paper, we provide the

amplitude and phase solutions for 1D loading of a uniform aquifer for a combination of hydraulic and mechanical loads, derived from first-principles. We make no claim of novelty in this regard, since the derivation is straightforward (though we have not found the complete solutions in the literature). The amplitude solution for pure hydraulic loading is well known and appears for example in G. van der Kamp and Maathuis (1991), as cited in our paper. We do, however, think it useful to bring the appropriate governing equations (neither over-simplified, nor unnecessarily generalised) to bear in the context of different possible loading conditions on the BAS in a clarifying manner. Remarkably, although poro-elastic theory is very well-established, it is new to fully apply it in the context of a very thick and extensive aquifer such as the BAS. Our paper determines for the first time the implications for groundwater pressurestogether with solid strains and displacements. We believe the implications are fundamental and widespread in the BAS, and very probably in other fluvio-deltaic environments. We suggest that the most important topics to focus on for the BAS are the implications for recharge and pumping under the most important generic condition present, i.e. widespread loading of the ground surface by water, for which a 1D analysis is suitable as explained in the paper (lines 55-56 in section 1 Introduction, and line 132 in section 2.1 Poromechanical equations). We have a companion paper in preparation which deals with the next level in the complexity hierarchy: the implications of poro-elasticity in the proximity of rivers, which requires analysis in 2D. We suggest these steps provide valuable insight and usefully prepare the way for a fully 3D analysis. As we point out in the paper, mechanical effects have not previously been taken into account in (vital) assessments of groundwater recharge (Shamsudduha et al., 2011), nor in regional models of groundwater flow (Mukherjee et al., 2007); neither have hydraulic effects been included in assessments of ground surface motions due to monsoonal loading (Steckler et al., 2010) – citations are provided in the paper. Significantly, our analysis supports the possibility that deep piezometers in the BAS may be suitable for 'geological lysimetry'; a potentially valuable method of measuring the terrestrial water budget locally that avoids the myriad assumptions of classical methods and the limitations of

remote-sensing. Thus, we would like to emphasise that the paper addresses an obvious 'gap' in the literature, in the bridging of which we have made a significant and original contribution. This would help to do a better estimate of recharge, particularly to the strategic deep groundwater, of the BAS to assess sustainability of groundwater development for irrigation, domestic and industrial uses.

POINT 2: "Due to using the commercial software, COMSOL, the results do not seem too realistic."

We have of course validated the numerical model versus our own analytical solutions; the match is so precise that there is nothing to discuss. We agree it makes sense to include this validation in Supplementary Information.

We would greatly appreciate if Reviewer#1 could identify any specific aspect of the simulations that (s)he considers in doubt.

POINT 3: "I expected to see some 3D models representing the deformation because of the loading."

We agree with the reviewer that there are conditions under which 2D and indeed 3D treatments of the mechanical and hydraulic responses will likely be important. Our next step is to provide a 2D analysis of the effects due to rivers (a manuscript is in preparation).

For the paper under review, a 1D examination is not only sensible in terms of starting at the base of complexity-hierarchy, but addresses a widespread and generically common condition in the BAS due to extensive flooding, rainfall events, and changes in unconfined groundwater storage. Apparently simple (yet physically plausible) areally-uniform loading conditions justify a 1D treatment, as described in the paper. We show these simple processes nonetheless lead to complex pressure and strain responses which are currently being misinterpreted by practitioners who neglect the coupling. We show that the dynamics due to different forms of loading (which may not be uniquely

apparent from a single groundwater hydrograph) can be decoded by taking into account the amplitude and phase effects; essentially we provide the basis for diagnosis of hydro-mechanical effects in 1D.

POINT 4: "Therefore, this paper is not a journal paper."

In view of the above, we rebut Reviewer#1's conclusion. The paper provides a fundamental re-appraisal of how groundwater hydrographs and surface displacements should be interpreted in the BAS and beyond, giving a diagnostic tool-kit which we demonstrate against real data. We use a well-established physics-solving platform, which we nonetheless validate for our governing equations against our own analytical solutions. We believe that the 1D treatment is particularly important in the BAS context due to spatially extensive surface water inundation and rainfall, and indeed it leads the way for future studies of hydro-mechanical effects at higher-dimensions.

---

## Referee Comment (RC2) · G. van der Kamp (Referee) · 14 Sep 2018

**General comments**

This paper addresses an important aspect of groundwater management: the fact that groundwater levels do not reflect only the changes of groundwater storage but also reflect groundwater pressure changes due to mechanical loading by changes of total water storage above the formation. For deep confined aquifers, such as the deeper layers of the Bengal Aquifer System, conventional estimates of groundwater recharge based on changes of groundwater levels may be in error if moisture loading effects are ignored. In turn recognition and analysis of such loading effects can provide valuable

constraints on the evaluation and simulation of surface water processes such as runoff and evapotranspiration. These loading effects are only starting to be acknowledged in groundwater management practice and this paper therefore can make an important contribution to the literature in groundwater and surface water hydrology. However, the paper can be considerably strengthened and focused and major revisions are recommended with regard to the review of poromechanical theory, the simulation of various loading scenarios, and the presentation and interpretation of field data from the Bengal Aquifer System..

**Specific comments**

The introduction provides an extensive general review of poro-elastic coupling and only gradually reaches the idea that changes in "terrestrial water storage" (TWS) result in changes of groundwater pressure. But the phenomenon of groundwater pressure changes due to changes of atmospheric pressure, as expressed through the concept of "barometric efficiency", is well-known in hydrogeology and is an obvious example of groundwater pressure changes caused by surface loading. It is only mentioned later in the paper (L 199) although it is most directly relevant to the focus of the paper.

The barometric efficiency of observation wells in deep confined aquifers also can used to determine an in situ value of specific storage that is directly applicable to the interpretation of short-term moisture loading events. Such analyses are briefly mentioned in the text (L 186-194) with reference to Burgess et al (2017), but are not further described or used in the paper although they are surely relevant. At the very least a more detailed explanation should be provided of why these results are not used.

L 102 onward – Poromechanical equations. This section starts off with a lengthy review of general 3D poro-elastic equations and then arrives back to the 1D differential equation that is used in the subsequent simulations and interpretations. This general review can be largely eliminated from the paper because it does not present anything new that cannot be found in the literature as cited. The paper could then perhaps go

directly to the 1D equation (# 9) including the discussion, more or less as give, n on when and why the 1D equations provide an adequate description of the poro-elastic interactions between stress and groundwater pressure. The appropriate equations for the loading efficiency and specific storage should be included - they are not given in the text as it stands.

L 148-359. The simulations of the three different loading scenarios can all be considered together as one, by treating the loading effects and the hydraulic head changes at the upper boundary separately. This approach is described and illustrated in detail by Anochikwa et al (2012) a reference that is important for this paper because it presents a somewhat similar analysis of poroelastic effects induced by moisture loading [Anochikwa, C.I., G. van der Kamp and S.L. Barbour, 2012. Interpreting pore-water pressure changes induced by water table fluctuations and mechanical loading due to soil moisture changes. Canadian Geotechnical J, 49(3): 357-366.]

With such an approach the Loading effects are manifested as pore pressure changes throughout the domain which then dissipate to the upper boundary, generally with very little and negligible effect on the hydraulic head at the upper boundary. Hydraulic head changes at the upper boundary propagate downward and dissipate. The two can be simply added to arrive at the total head changes throughout the domain. Such a decomposition is legitimate because it satisfies the boundary and domain conditions and the basic differential equations The advantage of such an approach to understanding and analyzing the combined effects of loading and flow is that the loading effects can be simulated and described separately from the effects of flow induced by changes of hydraulic head at the boundary. Such an approach may at first seem counter-intuitive, but it is mathematically and theoretically legitimate (given reasonably low compressibility of the formations), and can enhance understanding and visualization of the poroelastic processes.

L 229. Why ignore barometric effects? They can be easily dealt with by direct subtraction from the observation well records, and also provide a good estimate of loading

efficiency and compressibility of the formations. L 260. The assumption that loading efficiency is  $\sim$  1.0 is questionable and needs more justification, considering that the loading efficiency for barometric loading is an in-situ field measurement that closely corresponds in magnitude to the loading due to changes of TWS. Likely the instantaneous responses observed by the piezometers represent the short-term pore pressure changes in sands, whereas over the long term the pressure changes in clays may predominate. Thus the long-term value of specific storage, as applicable for changes of total water storage on an annual basis, may be  $\sim$ 1.0 since such a long-term response of the entire groundwater system may be dominated by the more compressible aquitard portions of the basin. This is an important and poorly resolved issue in geolysimetry and merits attention.

L 298-299. The "counter-intuitive" amplitude response to the LD is likely due to a "traveling wave" effect of the transient sinusoidal flows. In fact the flows for this case can be mathematically decomposed as the superposition of the imposed groundwater head changes due to loading (but without flow) and an equal but opposite sinusoidal transient imposed at the water table which induces a downward traveling wave that is dissipated as it moves, but may also be "reflected" from the horizontal boundaries represented by different hydraulic properties, thus giving rise to amplitude and phases that appear anomalous and counter-intuitive. LL 337-458 Field data. The reality of the loading effects due to changes of Total Water Storage could likely be demonstrated more strongly by including description and analysis of the short-term loading effects due to individual rain events. Such events are mentioned in the text and appear to be present in the hydrographs shown in figure 6 and especially in figure 7. The sharp spikes with subsequent decay that appear in the rising limbs of the hydrographs are presumably due to large rain events and subsequent runoff and evapotranspiration. Certainly such short-term responses to individual events should be apparent in the hydrographs if the hypothesis of water loading effects is at all correct.

There is no detailed discussion of the climate of the region and of whether seasonal

changes of total water storage of up to 1 meter, as implied by the records for the deep piezometers, are reasonable and realistic. The paper should therefore include a detailed presentation and discussion of the climate and water balance of the region, including estimates the water storage changes based on rainfall, runoff and evapotranspiration. L 449 the speculative uncertainty with regard to loading efficiency could perhaps be resolved by inspection of the responses at each depth to episodic rainfall events. As mentioned previously a description and analysis of barometric loading effects for the same piezometers would further establish the reality of the poroelastic responses to changes of total water storage.

**Technical corrections**

The reference information for Burgess et al (2017) is incomplete and requires more information as to the publisher and if necessary, how the report can be accessed.

LL 110-115 Can't have some units as Pa and others as MPa. That would require introduction of factors of 106 in the equations.

---

## Author Comment (AC2) · 18 Sep 2018

Response to comments by referee Garth van der Kamp

We are grateful to have these thoughtful and interesting comments from Dr. van der Kamp (GvdK), whose observations of surface moisture loading and subsequent development of geolysimetry as a technique with hydrological application have inspired our analysis applied to the Bengal Aquifer System.

We entirely agree that the issue of mechanical loading by changes in total water storage above the formation is an important aspect of groundwater management; our aim is to bring this to the attention of researchers and practitioners in the Bengal Aquifer System, and to those working in similar hydrogeological environments in the delta regions of SE Asia where groundwater meets the needs of over 300 million people. Suggestions that loading effects might be substantial in the Bengal delta were originally made by W.E. Bardsley and D.J. Campbell in 2000 (cited in the Discussion paper), yet the effects had not been recognised and documented prior to Burgess et al. (2017) in a paper preliminary to the present analysis.

Interesting questions certainly remain about the coupled hydro-mechanical behaviour of aquifers, which merit further attention, and we are grateful to GvdK for signalling these. The magnitude of 'loading efficiency' is one (para 6 of 'Specific Comments' in GvdK's review). The 'counter-intuitive' amplitude response for which GvdK suggests a 'travelling wave' explanation (para 7 of his 'Specific Comments' re L 298-299) is another. GvdK's comments on these points make interesting contributions which we reflect on below.

Rather than giving a linear point-by-point response, below we discuss the points under four titles: 'points to strengthen the paper', 'substantive criticisms', 'points of interest', and 'minor comments & technical corrections'.

Points to strengthen the paper:

The review has shed light on aspects of the paper which we would like to address in order to strengthen it:

- We agree that the phenomenon of groundwater pressure changes in response to changes of atmospheric pressure (the barometric effect) is a well-known concept and is a useful example of extensive surface loading (paras 1 and 2 of GvdK's'Specific Comments'). So we are grateful for the suggestion to add description of the barometric effect in the Introduction – this we will do.

- The comments about barometric effects and loading efficiency (para 6 of the 'Specific Comments') are well-taken. Our paper acknowledges in the Discussion (L 528-533) the interesting discrepancy been estimates of loading efficiency. We should like to give the point about time-scale of responses more prominence in section 2.4.1 of the paper, adding to our description of the relationship between specific storage, Young's modulus, and loading efficiency (L 186-204 of the paper).

- We agree that a summary of the climate and seasonality of the region in the introduction would give a useful context to our discussion of seasonal changes in terrestrial water storage (para 8 of 'Specific Comments'). Annual rainfall is around 2000 mm in this tropical monsoon climate, with individual events of >100 mm/day common during the monsoon season, May-November, during which river levels rise by 2-3 m leading to widespread flooding of the land surface. However we emphasise our aim in the paper under discussion is not to produce a site water balance, for which the data are not adequate. We agree that ultimately there is a need to quantify individual components of the water balance in a manner similar to Anochickwa et al. (2012), and for calibration of a deep piezometer to known changes in the water balance, but this was not our purpose in the paper under discussion. In Discussion at L 500 we write "A true indication of recharge requires ... a shallow tubewell screened over the depth interval of actual water table fluctuation". We should like to expand this statement to include the variety of measurements necessary to properly deconstruct the water balance using deep piezometers, such as has been done by Anochikwa et al (2012) at their site in Saskatchewan, Canada.

**Substantive criticisms:**

There are substantive criticisms made with respect to four points (headings in bold below):

Review of poromechanical theory (para 3 of 'Specific Comments', re L 102 onwards) It is suggested that the presentation of the poromechanical equations should be largely

СЗ

eliminated from the paper. We accept that our presentation of the poromechanical equations presents nothing new beyond the references cited, so might largely be eliminated. However, unless we transgress the length limit we think there is a good case for retaining the equations, thinking it beneficial to keep a more generalised treatment (despite this being available in the earlier literature). We have the following motivations:

(1) The equations provide a succinct reminder of the key assumptions, which although very well-known to those publishing poromechanical research, have been neglected elsewhere (e.g. it is assumed here that the skeleton grains are incompressible). The key insight (which has also been known for a long time), that we would like to reemphasise is that the 1D simplification leads to a partially coupled system. Moreover, the 1D simplification can then lead to decoupling if the change in total stress can be assumed insignificant. Perhaps it is because of the common applicability of the 1D simplification to groundwater flow that the great convenience of decoupling the flow equations has been so widespread. So, whilst elastic storage and barometric efficiency effects embed the poromechanical nature of aquifers, the decoupling simplification effectively removes the motivation to take loading into account when simulating flow and water storage. This is potentially seriously erroneous, particularly in the context of the BAS. We think it is so important that we have put it in the title, also at LL 57, 126 and extensively throughout the text, and return to it in the conclusions.

(2) In the event of groundwater flows to individual wells, loading by rivers and other more complex boundaries, the 1D assumption breaks down and full poromechanical coupling is likely to be needed. We have a manuscript in preparation which examines 2D effects where the geometric simplifications do not apply. We therefore prefer for the 1D paper not to start with unnecessarily approximated equations since we intend to provide a coherent body of work.

(3) Given the discussion about the operational meaning of barometric efficiency measurements (see below), we think it is doubly useful to have the fundamental equations to hand. Re: 'The appropriate equations for the loading efficiency and specific storage should be included - they are not given in the text as it stands.' (last sentence, para 3 of 'Specific Comments'): please see equations (5), (6) and (8) in the paper.

The upper boundary conditions of load and hydraulic head (paras 4 and 5 'Specific Comments', re LL 148-359) - which we use to represent the different scenarios of terrestrial water variation addressed in the paper. The reviewer recommends the approach used by Anochikwa et al (2012), and certainly this paper is highly relevant. A similar forensic approach to deconvolving the input terms is precisely what we in time intend for the Bengal Basin. We are aware of the Anochikwa et al. (2012) study; to omit it was an oversight – which we will correct.

We agree that Anochikwa et al.'s deconvolution of hydraulic and mechanical components is a valid way to solve the equations. But we contend that it is perfectly correct to solve the equations as we have done in the paper under discussion. The important aspect is that the boundary conditions are explicitly specified and compared. In that vein, we have noticed an interesting difference in assumptions between our work and the paper by Anochickwa et al: we add a mechanical load due to a moving water table (i.e. we include the weight of the water taken up as unconfined storage) whereas Anochickwa et al do not.

The Anochikwa et al. (2012) paper and ours under discussion have subtly different objectives: ours is to explore the different impacts of the specific styles of surface water load manifestation. We agree that the deconvolution approach is a valuable way to understand the water balance. However we are not clear quite how deconvolution can allow the scenarios we address to be treated "as one", since each overall scenario represents a distinct combination of boundary conditions. Also, the 'load-only' case requires a potentially misleading head boundary to be applied in order to obtain the correct superposition. Therefore, we would argue in favour of our approach, and our preference is to keep it.

Approach to treatment of the field data (para 7 of 'Specific Comments', re LL 337-458) The review suggests including description and analysis of the short-term rainfall events, as a strong demonstration of the reality of the loading effects. We completely agree with this suggestion - see L 375 for the Khulna site, "Episodic deflections on the hydrograph rising limbs, coincident with rainfall events, are likewise simultaneous at all measurement depths" and L 420 for the Laksmipur site, "The hydrographs are characterised by a sequence of episodic increments in groundwater head associated with periods of heavy rainfall". A previous paper by Burgess et al. (2017) showed the episodic increments in groundwater head at the Laksmipur site to be simultaneous with periods of rainfall, and proposed them as evidence for the loading response. However, we have not measured the individual components of the water balance at our sites in the paper under discussion, and cannot deconvolve their individual effects on the groundwater heads. Rather, we have tested the proposition that specific piezometers behave as geological weighing lysimeters (the approach is given at L 349-359), and for this purpose we have applied the appropriate piezometer head record as the upper boundary condition in the model, resolving "all sources of load acting at the site". Again, our purpose is subtly different to that of Anochikwa et al (2012).

Why ignore barometric effects? (para 6 of 'Specific Comments', re L 229) The justification for neglecting barometric effects on the generic simulations that we make is purely that of simplicity, as stated. It is straightforward to superpose a further loading signal on top of the existing one, but this would not bring further insights to the responses characteristic of particular surface boundary conditions, which is our purpose. In terms of the simulations of specific sites, the daily perturbation on water heads by atmospheric pressure changes in the data is of the order of 1cm, which is relatively small compared to the annual perturbation of the order of 1m. It would need to be taken into account when deconvolving deep piezometric signals to make water resources assessments, particularly to include the seasonal atmospheric pressure variations, but we don't make such assessments in the paper under discussion. We would like to add such an explanation to the discussion section of the paper, where we try to move from 'lessons learned' to how methods might be applied. For the real site simulations the point is that the top boundary is given a total load which is the sum of atmospheric and water loads (see L 383 of the paper: "The upper boundary resolves all sources of load acting at the site including from the Rupsa River, which is a linear rather than an areally-extensive load". Our objective was (LL 341-344) to "apply the principles and assumptions of the partially-coupled hydro-mechanical approach to reproduce the characteristic features of the multi-level groundwater hydrographs ...., rather than to attempt an exact match by inverse modelling". Therefore our objectives were subtly different from Anochikwa et al (2012). To obtain a proper water balance would indeed require removal of the barometric contribution.

Evaluation of loading efficiency, and use of barometric efficiency (paras 2 and 6, re L 260, of 'Specific Comments') For the generic simulations, making LE=1 neatens the analysis. It could easily have been changed, say to LE=0.8, but this would provide no additional insight.

The reviewer comments: 'Such analyses (of barometric efficiency) are briefly mentioned in the text (L 186-194) ...., but are not further described or used in the paper although they are surely relevant. At the very least a more detailed explanation should be provided of why these results are not used.' In the paper we explain why we did not use barometric efficiency as our basis for loading efficiency – see LL 186-204 in the paper, which concludes "Therefore for the purposes of this paper we adopt Ss estimates based on field measurements and use the corresponding  $\delta IZ_i$  and E values." Later, in the Discussion, we return to this as a topic which requires further attention in agreement with the reviewer's point– see LL 528-533: "In our analysis we have based values for the 3D loading efficiency,  $\beta$  (0.961-0.996) and Young's Modulus, E (82-851 MPa) in the BAS on field measurements of Ss, for the sake of internal hydro-mechanical consistency, but we have noted a discrepancy with lower values for the 1D loading efficiency  $\delta IIJL'$  (0.69-0.87) derived from determinations of barometric efficiency(Burgess et al., 2017). These differences require attention, but the overall conclusions on the significance of poroelastic behaviour in the BAS and the pattern of poroelastic responses characteristic of specific upper surface TWS boundary conditions are unaffected."

Therefore, we completely agree with the reviewer's important point that barometric efficiency measurements operationally consider timescales corresponding to short-term periods governed by the pore pressure changes in the relatively stiff sediments (para 6 of 'Specific Comments', re L 260). In Burgess et al (2017) short-term moisture loading effects were a key interest, so loading efficiencies based on barometric efficiency estimation are appropriate. However, in the paper under discussion we are concerned over poromechanical consistency, and contend that we should remain sceptical about using barometric efficiencies derived from short-term responses to address water load changes operating over the longer term. Interestingly, longer-term loads are potentially more readily determined in the Bengal Basin than in drier, temperate environments, since there is the possibility of measuring time-series of flood inundation over the monsoon season, rather than individual components of the soil/vegetation water budget. Therefore we also agree with the reviewer that this is an important and poorly resolved issue – see L 528-533 in the Discussion section of the paper, and below under 'points of interest'. We can augment our discussion of this point in the paper.

Points of interest: 'loading efficiency' and the 'counter-intuitive' amplitude Loading efficiency We felt that the discrepancy that we haven't resolved between field measurements of Ss (via pumping tests), anticipated material stiffness and barometric efficiencies is sufficiently interesting that we have made it prominent. We agree with the reviewer on this – also see Discussion L 528-533.

The 'counter-intuitive' amplitude response to the 'load only' upper boundary scenario: The reviewer makes a very interesting point here (para 7 of 'Specific Comments, re L 298-299), and to build on it we would like to consider adding a paragraph to the Discussion section of the paper. The decomposition described is a helpful way to mathematically picture how the apparently anomalous amplitude and phases come about in the 'load only' case. We think our partially-coupled solution is also useful, however. For example, for the 'load-only' case one can take the phase solution we give in equation [12] and set  $\alpha$ =0 and  $\gamma$ =1; in the asymptotic limit as z tends to 0, the phase tends to  $-\pi/4$ . By differentiating we get the dimensionless depth corresponding to the peak amplitude, i.e. the solution to  $\cos(\theta)$ + $\sin(\theta)$ =exp(- $\theta$ ) which is ~2.284 and the peak amplitude at this depth is ~1.07. So, we would argue that both approaches can be useful in different ways.

Minor comments and technical corrections The reviewer makes some minor comments which we address as follows:

- 'The appropriate equations for the loading efficiency and specific storage should be included - they are not given in the text as it stands' (para 3 of 'Specific Comments' in the review) – please see equations (5), (6) and (8) in the paper.

- We will give full details for Burgess et al (2017), published in Scientific Reports, 7(1), 3872. doi:10.1038/s41598-017-04159-w

- Units as Pa and MPa (final point in GvdK's review, re LL 110-115): Yes, we accept – but emphasise that all the same units were included in application of equation (1) so no corrections to our working are needed.

---

## Author Comment (AC3) · 18 Oct 2018

Authors' summary response to HESS editor.

Our thanks for your patience; the first and second authors have both been out of the country since receiving the last of the reviewers' comments.

Reviewer#1

We have addressed each of Reviewer#1's comments directly in our original reply (hess-2018-304-AC1). There we point out that this is the first paper that we are aware of to apply partially coupled poro-mechanical models to a deep and extensive sedimentary

sequence such as the Bengal Aquifer System, which 150 million people rely on for water. We deal with the simplifications that are appropriate to areally-extensive loading types which are common and important in the context of the BAS and therefore have generic impact on groundwater hydrographs and surface displacements throughout the basin. In the article we provide a methodology for identifying key boundary conditions from hydrograph data. We demonstrate numerically for the first time by reference to the Bengal Basin (i) the limits to the conventional use of water levels in piezometers to indicate groundwater recharge and drainage (ii) the conditions under which piezometer water levels respond primarily to changes in terrestrial water storage mass, so may be suitable for 'geological lysimetry', and (iii) the likely scale of ground surface vertical deflections, to complement GPS studies of ground surface motion. We entirely agree with the reviewer that there are also circumstances under which poro-mechanical interactions will need to be addressed in 2D and 3D; these are the subjects of companion papers that will follow. We contend therefore that the paper addresses an obvious 'gap' in the literature, and makes a significant and original contribution. The improvements we have suggested, based on both reviews, we hope will help readers (including Reviewer#1) engage better with the paper by making its contribution and novelty clearer.

**Reviewer#2**

Reviewer#2 (Dr. Garth van der Kamp) headlined his thoughtful and constructive comments with the judgement that "This paper addresses an important aspect of groundwater management: the fact that groundwater levels do not reflect only the changes of groundwater storage but also reflect groundwater pressure changes due to mechanical loading by changes of total water storage above the formation", and also that "These ... effects are only starting to be acknowledged in groundwater management practice and this paper therefore can make an important contribution to the literature in groundwater and surface water hydrology." He went on to raise some interesting points and to suggest ways to strengthen the paper.

We concentrate below on a summary of our response to the review of substance provided by Reviewer#2. The arrangement of the points and the full reasoning are as given in our original author response (hess-2018-304-AC2). Included in this summary are the modifications we propose in the light of Reviewer#2's review, and which we agree will strengthen the paper.

Points proposed to strengthen the paper - We will add to the Introduction (section 1 of the paper) a description of the barometric effect as a useful example of extensive surface loading. - We will augment section 2.4.1 of the paper, expanding on Reviewer#2's point about time-scale of responses in relation to barometric effects and estimation of loading efficiency. - We will add to the Introduction a summary of the climate and seasonality of the region, as context for the subsequent discussion of seasonal changes in terrestrial water storage. - We will include citation of Anochikwa et al (2012) at L 69 in the Introduction, which we omitted in an oversight; it is an important example of geological weighing lysimetry as applied in Saskatchewan, Canada. Also, at L 500 in the Discussion (section 5.2 of the paper) we will expand on the variety of measurements necessary to properly deconstruct the water balance using deep piezometers, such as has been done by Anochikwa et al (2012).

Responses to substantive criticisms - We should like to retain our presentation of the poro-mechanical equations (Re para 3 of Reviewer#2's 'Specific Comments' in which he suggests largely eliminating them), motivated by the three reasons given in our original response to Reviewer#2's comments. (Please note that the appropriate equations for the loading efficiency and specific storage requested by Reviewer#2 are included in the original Discussion paper, see equations 5, 6 and 8). - Re the representation of upper boundary conditions of load and hydraulic head (paras 4 and 5 of Reviewer#2's 'Specific Comments'), we agree that Anochikwa et al.'s (2012) deconvolution of hydraulic and mechanical components is completely valid, but we nonetheless contend that the approach we use in the Discussion paper to solving the equations is also perfectly correct. In the paper, we do not repeat the analysis/discussion over whether hydraulic influences alone may explain deep hydrograph data in the BAS since this

СЗ

possibility has been comprehensively dealt with in Burgess et al. (2017). We agree it is nonetheless useful to include, so we have done as suggested by Reviewer#2 and addressed each boundary condition separately (in this case the hydraulic boundary, the load boundary, and the pumping effects) in plots such as those shown in Anochikwa et al (2012), which would be suitable for inclusion as 'Supplementary Information'. For the main text, we prefer to retain our approach, for the reasons given in our original author response, and emphasise that our objective is subtly different from that of Anochikwa et al. (2012): ours is to explore the different impacts of the specific styles of surface water load manifestation; Anochikwa et al's is to deconstruct the water balance. Regarding the 'counter-intuitive' amplitude response to the 'load only' upper boundary scenario (para 7 of Reviewer#2's 'Specific Comments'), we would like to add consideration of this interesting point to section 5.1 in the Discussion section of the paper, following the outline given in our original author response. - In our approach to treatment of the field data (para 7 of Reviewer#2's 'Specific Comments'), we do not examine individual short-term rainfall events, although we do comment on rainfall effects at each site, as referenced in our original response to Reviewer#2's comments and in our citation of Burgess et al (2017) for the effects at one of the sites. Emphasising again that our purpose is subtly different to that of Anochikwa et al (2012), we have not measured the individual components of the water balance at our sites in the paper under discussion. so cannot resolve their individual effects on the groundwater heads. Rather, we have tested the proposition that specific piezometers behave as geological weighing lysimeters (the approach is given at L 349-359 in section 4 of the paper 'Applying the partial coupling analysis to field data'), and for this purpose we have applied the appropriate piezometer head record as the upper boundary condition in the model, resolving "all sources of load acting at the site", including barometric loading. - Re "Evaluation of loading efficiency, and use of barometric efficiency" (paras 2 and 6 of Reviewer#2's 'Specific Comments'), there are 3 key issues raised by Reviewer#2 relating to barometric efficiency: (i) we don't include barometric effects in the generic modelling. Response: this is for simplicity, since although it is trivial to include, its inclusion would

serve to de-clarify the analysis. (ii) we don't use the barometric efficiency estimates (although they are available from Burgess et al, 2017) to estimate loading efficiency. Response: please see references to section 2.4.1 LL 186-204 and to the Discussion LL 528-533 in our original author response. In the Discussion we have highlighted the inconsistency between estimates of mechanical stiffness, storativity and barometric efficiencies. As Reviewer#2 points out, barometric efficiency estimates may have a time-dependency due to pressure diffusion in aquitard layers. We will add these points to section 2.4.1 of the paper as a more detailed explanation, as suggested by Reviwer#2. We agree that this remains an "important and poorly resolved issue in geolysimetry". In our Discussion we put it thus: "These differences require attention, but the overall conclusions on the significance of poroelastic behaviour in the BAS and the pattern of poroelastic responses characteristic of specific upper surface TWS boundary conditions are unaffected". (iii) we include the barometric load as part of the total load that we apply to the surface in the field examples, without attempting to de-convolve each contribution. Response: we do this since we do not have shallow water data nor rainfall measured precisely at the field sites, and it is not our objective to deconstruct the water balance (see also the two previous bullet-points). We will add the explanation from our original author response to the discussion section of the paper, where we move from 'lessons learned' to how methods might be applied.

Minor comments and technical corrections For the three points made by Reviewer#2, please see our original author response in which all are readily addressed.

Concluding statement We acknowledge the thoughtful comments made by Reviewer#2, and we are confident that the modifications we have proposed above will strengthen the paper. We would be grateful for this opportunity.

---

## Author Response (AR1)

**A partially-coupled hydro-mechanical analysis of the Bengal Aquifer System under hydrological loading**

Nicholas D. Woodman, William G. Burgess, Kazi Matin Ahmed and Anwar Zahid

**Text modifications in response to directions by the Editor**
[Please also see Author summary responses to Editor, on the Discussion webpage]

**Editor directions are in bold.**
Responses are in plain script. Line numbers refer to re-submitted ms showing tracked changes unless otherwise stated.
*Added text is given in italics.*

1. **The authors should highlight the novelty (and the potential impact) of their work**

Response: To emphasise the novelty of the paper (also requested by Reviewer#1), the following text has been added to the to the Introduction at L86:
*'Poro-elastic theory is very well-established, but has not previously been applied in the context of a thick and extensive aquifer such as the BAS to show the implications for groundwater pressures together with solid strains and ground surface displacements.'*

Please also note that the four sub-sections of the Discussion each concentrate on a separate aspect of the impact of the research.

2. **Section 2.1: Rev 2 suggested to shorten significantly this session, since it does not include any novel element. I tend to agree with Rev 2. In order to facilitate the reading of the manuscript I suggest to move this section to an Appendix (or to include it as supplementary information)**
**- Barometric effects. Add in the manuscript why barometric effects have been neglected (or include them, as suggested by rev 2)**

Response: Please see response to point (3) of Reviewer#2. In the re-submitted ms we have made a substantial reduction of Section 2.1 (Poromechanical equations), and have removed the standard derivations to an Appendix, as suggested, thereby replacing L103-137 of the original manuscript by an abbreviated text (not replicated here, but see the tracked-changes ms). Equation references have been renumbered accordingly throughout.

3. **check your manuscript carefully for typos, …. , terminology, updates of data ……, or updates of variables in equations.**

Response: We have made a few typo corrections or minor edits for clarity; please see track-changed ms. This includes some new text inserted at L29 where the following text is added:
*"More than 10 million tubewells throughout the basin provide water from BAS for domestic use and for irrigation of the rice crop (Ravenscroft et al., 2009); these include hand-pumped tubewells, normally between 15 and 30 m depth below ground level (bgl), for domestic use, and tubewells installed with motor-driven pumps to abstract water from between 50 and 75 m depth bgl for irrigation of the dry season rice crop (January to April). Municipal water supplies commonly abstract year-round from depths between 200 and 300 m bgl (Shamsudduha et al., 2018)."*

**Text modifications in response to comments by Reviewer#1**
[Please also see responses and rebuttals made as Author responses to Reviewer#1's comments.]

**Review comments are in bold.**
Responses are in plain script.
*Added text is given in italics.*

**I don't see any new method or development. The only new thing I can see here is just the application, which I think is not enough.**

Response: In emphasis of the novelty of the paper (also requested by the Editor), the following text has been added to the to the Introduction at L86 in the re-submitted ms:
 *'Poro-elastic theory is very well-established, but has not previously been applied in the context of a thick and extensive aquifer such as the BAS to show the implications for groundwater pressures together with solid strains and ground surface displacements.'*

**Text modifications in response to comments by reviewer#2, Garth van der Kamp**

[Please also see responses and rebuttals made as Author responses to Reviewer#2 comments.]

**Review comments are in bold.**
Responses are in plain script. Line numbers refer to re-submitted ms showing tracked changes unless otherwise stated.
*Added text is given in italics.*

Specific comments

**1. (In the Introduction) … the phenomenon of groundwater pressure changes due to changes of atmospheric pressure, as expressed through the concept of "barometric efficiency", is well-known in hydrogeology and is an obvious example of groundwater pressure changes caused by surface loading. It is only mentioned later in the paper (L 199) although it is most directly relevant to the focus of the paper.**

Response: In the re-submitted ms, in an addition to the Introduction, we now include the barometric effect as a prime example of surface loading. At L56 the following text is added:
*'Furthermore, it is associated with the well known concept of barometric efficiency (Spane et al., 2002), which describes the response of groundwater pressure to variations in atmospheric pressure, perhaps the example of surface loading effects most familiar to hydrogeologists.'*

**2. The barometric efficiency of observation wells in deep confined aquifers … are briefly mentioned in the text (L 186-194) with reference to Burgess et al (2017), but are not further described or used in the paper although they are surely relevant. At the very least a more detailed explanation should be provided of why these results are not used.**

Response: In the re-submitted ms, the following text has been added at L251:
*'The discrepancy may alternatively be related to the timescale of processes responsible for changes in groundwater pressure. Barometric efficiency measurements operationally consider short-term pore pressure changes likely corresponding to the response of relatively stiff aquifer sands, whereas pressure changes in clays are expected to become significant in the longer term. Where short-term moisture loading effects are the key interest (Anochikwa et al. 2012, Bardsley and Campbell 2000), values for loading efficiency derived from barometric efficiency may be the most appropriate. Here however our main concern is for poromechanical consistency and for water load changes operating over a range of time scales, therefore we adopt Ss estimates based on field measurements and use the corresponding β and E values (Figure 3).'*

**3. L102 onwards - Poromechanical equations. This section starts off with a lengthy review of general 3D poro-elastic equations and then arrives back to the 1D differential equation that is used in the subsequent simulations and interpretations. This general review can be largely eliminated from the paper because it does not present anything new that cannot be found in the literature as cited. The paper could then perhaps go directly to the 1D equation (# 9) including the discussion, more or less as given on when and why the 1D equations provide an adequate description of the poro-elastic interactions between stress and groundwater pressure.**

Response: In the re-submitted ms we have made a substantial reduction of Section 2.1 (Poromechanical equations), and have removed the standard derivations to an Appendix, as suggested by the Editor, thereby replacing L103-137 of the original manuscript by an abbreviated text (not replicated here, but see the tracked-changes ms). Equation references have been renumbered accordingly throughout.

**4. The appropriate equations for the loading efficiency and specific storage should be included - they are not given in the text as it stands.**

Response: In the re-submitted ms the equations are given at L168-172.

**5. L 148-359. The simulations of the three different loading scenarios can all be considered together as one, by treating the loading effects and the hydraulic head changes at the upper boundary separately. This approach is described and illustrated in detail by Anochikwa et al (2012) a reference that is important for this paper because it presents a somewhat similar analysis of poroelastic effects induced by moisture loading.**

Response: Regarding the possible alternative manner of demonstrating the coupled hydro-mechanical effects of surface water loading, individually and collectively in the manner of Anochikwa et al. (2012), in the re-submitted ms we have drawn a comparison between our study and that of Anochikwa et al (2012) in the following text added at L554:
*'Anochikwa et al. (2012) assembled field measurements of rainfall and evapotranspiration at a site in Saskatchewan, Canada, using them to define the upper boundary conditions in a one-dimensional model to examine their hydraulic and mechanical loading separately, before summing the outcomes to simulate the overall hydro-mechanical influence on groundwater pressure. Having determined loading efficiency by reference to barometric effects, they then calibrated their 1D model against observed groundwater pressures by varying hydraulic conductivity. At Khulna and Laksmipur, measurements of the separate components of the terrestrial water cycle were not available, hence an indirect demonstration of hydro-mechanical effects was desirable.'*

In addition, we have included in the Supporting Information a decomposition analysis such as suggested by Reviewer#2 and in the manner of Anochikwa et al (2012), applied to the field data of groundwater levels at Laksmipur, south-east Bangladesh. We have added the following sentence at L370 of the ms:
*'The 'LD' behaviour can be interpreted by means of a decomposition of heads in the manner shown in Anochickwa et al. (2012) (see Supporting Information).'*

Further reference to the very relevant paper by Anochikwa et al. (2012) has also been added at lines 72, 199, 255, 619.

**6. L 229. Why ignore barometric effects? They can be easily dealt with by direct subtraction from the observation well records, and also provide a good estimate of loading efficiency and compressibility of the formations.**

Response: In the re-submitted ms the following sentence is added at L282:
*'The daily perturbation on water heads by atmospheric pressure changes is of the order of 0.01m (Burgess et al. 2017), which is small compared to the annual hydrograph amplitude of the order of 1 m. Barometric pressure and earth tides are both neglected for simplicity here.'*

Also, in the Discussion section 5.4 of the re-submitted ms, we have added the following sentence at L617:
*'Although we omitted barometric effects in the generic simulations for the sake of simplicity, it is straightforward to superpose a further loading signal on top of the existing one if required, as for example when deconvolving deep piezometric signals to make water resources assessments (Anochikwa et al. 2012).'*

**7. L 260. The assumption that loading efficiency is ~1.0 is questionable and needs more justification, considering that the loading efficiency for barometric loading is an in-situ field measurement that closely corresponds in magnitude to the loading due to changes of TWS. … This is an important and poorly resolved issue in geolysimetry and merits attention.**

Response: Please see response to point 2 above for our justification.

**8. L 298-299. The "counter-intuitive" amplitude response to the LD is likely due to a "traveling wave" effect of the transient sinusoidal flows. In fact the flows for this case can be mathematically decomposed as the superposition of the imposed groundwater head changes due to loading (but without flow) and an equal but opposite sinusoidal transient imposed at the water table which induces a downward traveling wave that is dissipated as it moves, but may also be "reflected" from the horizontal boundaries represented by different hydraulic properties, thus giving rise to amplitude and phases that appear anomalous and counter-intuitive.**

Response: We agree that decomposition of the solutions is a helpful way to mathematically picture how the apparently anomalous amplitude and phases come about in the 'load only' case. In the re-submitted ms we have deleted the words 'counter intuitive' at L362, and have added the following sentence at L370:
*'The 'LD' behaviour can be interpreted by means of a decomposition of heads in the manner shown in Anochikwa et al. (2012) (see Supporting Information).'*

**9. LL 337-458 Field data. The reality of the loading effects due to changes of Total Water Storage could likely be demonstrated more strongly by including description and analysis of the short-term loading effects due to individual rain events. Such events are mentioned in the text and appear to be present in the hydrographs shown in figure 6 and especially in figure 7. The sharp spikes with subsequent decay that appear in the rising limbs of the hydrographs are presumably due to large rain events and subsequent runoff and evapotranspiration. Certainly such short-term responses to individual events should be apparent in the hydrographs if the hypothesis of water loading effects is at all correct.**

Response: Short-term responses to individual rain events at both field sites are acknowledged (see L386 of the Discussion paper for the Khulna site, "*Episodic deflections on the hydrograph rising limbs, coincident with rainfall events, are likewise simultaneousat all measurement depths*", and L 431 of the Discussion paper for the Laksmipur site, "*The hydrographs are characterised by a sequence of episodic increments in groundwater head associated with periods of heavy rainfall*". However, we have not measured other components of the water balance at the sites in the same manner as Anochikwa et al (2012), and therefore cannot deconvolve their individual effects on the groundwater heads. Rather, we have tested the proposition that specific piezometers behave as geological weighing lysimeters (the approach is given at L 349-359 of the Discussion paper), and for this purpose we have applied the appropriate piezometer head record as the upper boundary condition in the model, resolving "*all sources of load acting at the site*".

**10. There is no detailed discussion of the climate of the region and of whether seasonal changes of total water storage of up to 1 meter, as implied by the records for the deep piezometers, are reasonable and realistic.**

Response: In the re-submitted ms, the following text has been added at L90:
*'The Bengal Basin has a tropical climate dominated by the Indian monsoon, with annual rainfall increasing from 1500 mm in the south and west to 5500 mm in north-east Bangladesh, of which 85% falls during the summer rainy season (May to November) when individual storm events can*

*contribute over 100 mm per day (Ravenscroft, 2003). During the monsoon season, river levels rise by 2-8 m leading to widespread flooding (Steckler et al., 2010) with up to 30% of the land surface routinely being flooded to a depth up to ca. one metre. During the Boro rice irrigation season (January to April), groundwater pumping for irrigation throughout rural areas commonly provides standing water across rice paddies to a depth of ca 0.1 m (Hasanuzzaman, 2003).'*

**11. L 449 the speculative uncertainty with regard to loading efficiency could perhaps be resolved by inspection of the responses at each depth to episodic rainfall events. As mentioned previously a description and analysis of barometric loading effects for the same piezometers would further establish the reality of the poroelastic responses to changes of total water storage.**

Response: Please see Responses to points 9 and 2.

Technical corrections

**The reference information for Burgess et al (2017) is incomplete and requires more information** ….
Response: The publication details are now complete.

**LL 110-115 Can't have some units as Pa and others as MPa. That would require introduction of factors of $10^6$ in the equations.**
Response: We confirm that identical units were included in application of equation (1) so no corrections to our working are needed.

[revised manuscript text omitted]

25, 2035-2059, 2011.

---

## Author Response (AR2)

Table of minor revisions and replies to reviewer

Nick Woodman et al., 22 March 2019

| Reviewer comment | Reply | Change |
|---|---|---|
| The conclusions of the paper should include a recommendation to the effect that both water table data and deeper head data should be obtained to support analysis of the groundwater dynamics of large layered aquifers, such as the BAS. | Agree - Done | L364 "Data on changes of the actual water table at the field sites are unfortunately not available."

L573 "Data on changes of the water table would have greatly helped the analysis of loading effects at Khulna and Laksmipur. It is strongly recommended that in future hydro-mechanical analyses of the groundwater dynamics of large layered aquifers such as the BAS, both water table data and deeper head data should be obtained. For the water table, this requires a shallow piezometer to be screened across the full range of fluctuation of the true water table. " |
| Line 133-137. Equations for the one-dimensional specific storage coefficient and loading efficiency are given in the discussion following equation 1and are formulated in terms of 3D poroelastic equations. This formulation is not incorrect, but it is needlessly complex and is likely to confuse many readers, who may be left wondering whether these two parameters, as defined in this paper, are the same as the familiar parameters which go by that name in established groundwater texts and literature. It is preferable to present the equations for these two parameters in the conventional form of standard groundwater theory, well-known and accepted since Jacob (1940). How these are derived from the general 3-dimensional poro-elastic equations can be left to the appendix and has in any case been shown in previous literature. | Agree - Done | L133 definitions removed to the appendix. |
| Line 165. The hydraulic diffusivity is in fact just κ /Ss and κ (hydraulic conductivity) has already been introduced in equation 1. It is a needless complication to now introduce the factor $k/\rho g \mu$ without even mentioning that $k\rho g \mu =$ κ. Therefore $k/\rho g \mu$ should be replaced by κ. | Agree – this was left-over from the previous change | $D = \frac{k}{S_s}$. |

| | | |
|---|---|---|
| L 208-213 (Fig. 3). This figure presents the 3D loading efficiency, but the analysis and discussion of the field data are in terms of the 1D loading efficiency. Therefore Fig 3 would be more relevant and useful to the analysis and discussion if it were to present the 1D loading efficiency instead. | Agree.

(on double-checking we find this was indeed 1D loading efficiency, but mislabelled as $\beta$ – now corrected) | The Y-axis label and the caption for Figure 3 have been amended accordingly. |
| L 224. The reference to "field methods" here appears to refer to the high-pressure dilatometer measurements at Padma bridge (Da Silva et al), measurements that presumably were short-term and carried out with stress changes that are far larger than the ambient stress changes due to barometric pressure changes and TWS changes. It is not clear why the dilatometer results, which suggest a much more compressible material, would be more appropriate to use for analysis of the TWS effects. | The reference was intended to mean aquifer pumping tests.

We just wanted to point out that the pressuremeter data appear to corroborate / be consistent with the stiffnesses that we estimated in the paper via storativity values. But it is worth remarking in addition, that unload-reload stiffnesses are generally higher than for virgin loading and less affected by strain, so unless there is another form of bias the measurement would not be expected to be a significant underestimate of the stiffness over a smaller strain range. | 'field measurements' replaced with 'aquifer pumping tests'.
'confirmed' changed to 'corroborated' |
| At the very least it would be instructive to carry out the simulations with a range of values for E (and Ss and 1D loading efficiency) that reflects both the barometric and dilatometer field results. | In equations (5) and (6), $\gamma = S_y \xi$. For our plots, $\xi = 1$, so $\gamma = S_y$ and the legend could be replaced with $\gamma$.

Thus, the effect of changing $\xi$ is the same as for changing Sy. | The following has been added to the caption for Figure 5:
(Note, in the instance that $\xi$ is not close to 1, $S_y$ in these plots can be substituted by $\gamma = S_y \xi$) |
| L 413-416 It is questionable to extrapolate the 0.06 ratio for daily (tidal) variations to the annual variation. The response of the piezometer to the longer-term annual changes of the Rupsa River is likely to be much greater and may well be close to the annual change of the river stage, depending on the hydraulic connections between the aquifer and the river. Therefore the assumption that piezometer | We agree that we must be careful in reducing a 2D/3D effect to a 1D simplification. We have made a modification to the script which acknowledges and emphasizes | The following text is added in Section 4.1:
The ratio of daily (tidal) variability in head at KhPZ60 and in the Rupsa River level is ~0.06. At an equivalent loading efficiency, the |

| | | |
|---|---|---|
| KhPZ60 reflects a pure TWS signal is questionable. This assumption also implies that the surface moisture load changes by about 2 m annually, which seems an unrealistically large change unless much of the area is flooded by the end of the monsoon season. A much more sophisticated 2D or 3D simulation is likely required, as pointed out in other parts of the paper and as should be clearly stated in the discussion of the Khulna piezometer data. | the caution the reviewer is expressing here. This addition is consistent with what we already say in Section 5.4 'Limitations' (at L577 of the ms as reviewed) "Under certain circumstances the extensive load assumption inherent in the 1D analysis may break down. Rivers, as linear sources of head and load, can be accommodated within the 1D framework where their contribution to the TWS load is minor as demonstrated at Khulna. In general however, rivers should be expected to impose laterally variable heads and require a more generalised 2D or 3D fully-coupled poro-mechanical treatment (Boutt, 2010;Pacheco and Fallico, 2015"); and also in the Conclusions (at L590 of the ms as reviewed) "Rivers can be incorporated as a component of the 1D load where their contribution is small, but in general will require a 2D or fully 3D treatment." | 1.23 m annual variation in river stage would explain ~0.07 m head variation in KhPZ60, only 3% of the total. While the response of KhPZ60 to the annual changes of the Rupsa River may be greater than to the tidal changes, depending on the details of aquifer structure and hydraulic connection to the river, 97% of the annual variation in head at the piezometer is taken here as attributable to changes in TWS other than load transmitted from the river, representing areally-extensive loads as required by the 1D partially-coupled analysis. This is likely an over-estimate; measurements of true water table fluctuation and surface flooding depths in the vicinity are necessary to constrain the hydro-mechanical model more closely. |
| L 477 The finding that "For LkPZ244 the simulated heads are an excellent match with measurements over the entire period" is partly due to the fact that the measurements for LkPZ244 were used as TWS input for the simulation. The excellent match reflects the choices of Kv and Ss values such that changes of head at the upper boundary do not penetrate to the depth of LkPZ244, as also suggested by the simulation results presented in Fig 4c. But it is not entirely obvious that this perfect match would obtain for all values of Kv and Ss within reasonable ranges for the BAS. | This is true. We don't show that the results are valid for all conditions at but take a simple set of parameters. The purpose of the simulation is to show that a very simple model, informed by local measurements may be sufficient to explain the data, not to show that geolysimetry is generically possible under all conditions. | No change. |

[revised manuscript text omitted]

25, 2035-2059, 2011.